# Astrocytic chloride is brain state dependent and modulates inhibitory neurotransmission in mice

Verena Untiet [1], Felix R. M. Beinlich [1], Peter Kusk[1], Ning Kang[2], Antonio Ladrón-de-Guevara [2,3], Wei Song[2], Celia Kjaerby [1], Mie Andersen [1], Natalie Hauglund [1], Zuzanna Bojarowska [1], Björn Sigurdsson [1], Saiyue Deng[4], Hajime Hirase [1], Nicolas C. Petersen [1], Alexei Verkhratsky [1,5,6] & Maiken Nedergaard [1,2]

Information transfer within neuronal circuits depends on the balance and recurrent activity of excitatory and inhibitory neurotransmission. Chloride ($Cl^-$) is the major central nervous system (CNS) anion mediating inhibitory neurotransmission. Astrocytes are key homoeostatic glial cells populating the CNS, although the role of these cells in regulating excitatory-inhibitory balance remains unexplored. Here we show that astrocytes act as a dynamic $Cl^-$ reservoir regulating $Cl^-$ homoeostasis in the CNS. We found that intracellular chloride concentration ($[Cl^-]_i$) in astrocytes is high and stable during sleep. In awake mice astrocytic $[Cl^-]_i$ is lower and exhibits large fluctuation in response to both sensory input and motor activity. Optogenetic manipulation of astrocytic $[Cl^-]_i$ directly modulates neuronal activity during locomotion or whisker stimulation. Astrocytes thus serve as a dynamic source of extracellular $Cl^-$ available for GABAergic transmission in awake mice, which represents a mechanism for modulation of the inhibitory tone during sustained neuronal activity.

Maintaining the balance between excitation and inhibition within a narrow range is critical for proper brain function. To sustain the activity of neuronal networks in time and space, glutamatergic excitation is counteracted by the GABAergic inhibition. Neuronal activity in different brain states is controlled by neurotransmitters, neuromodulators and non-synaptic mechanisms operating at a slower timescale[1–3]. The ionic composition of the extracellular space is a powerful regulator of brain function: changes in extracellular ion concentration can bypass neuromodulator signalling and directly alter EEG and behavioural state[4,5]. Cortex-wide increases in extracellular $K^+$ concentration ($[K^+]_o$) accompany transition from quiet wakefulness to locomotion, while artificial elevation of $[K^+]_o$ enhances spiking and improves motor performance[6]. Sleep duration and sleep architecture are profoundly influenced by fluctuations in $[K^+]_o$[7–10] reflecting the important role of $[K^+]_o$ in modulation of neuronal membrane potential, excitability, spiking and oscillatory activity[11–20]. During sleep a decrease in $[K^+]_o$ is paralleled by an increase of $[Ca^{2+}]_o$, $[Mg^{2+}]_o$, $[H^+]_o$, and the volume of the interstitial space[4]. Brain state-dependent regulation of extracellular anions remains, however, unknown. Inhibitory transmission is mediated by $Cl^-$ fluxes driven by the electrochemical

[1]Division of Glial Disease and Therapeutics, Center for Translational Neuromedicine, University of Copenhagen, 2200 Copenhagen, Denmark. [2]Division of Glial Disease and Therapeutics, Center for Translational Neuromedicine, Department of Neurosurgery, University of Rochester Medical Center, Rochester, NY 14642, USA. [3]Department of Biomedical Engineering, University of Rochester, Rochester, NY 14627, USA. [4]Department of Neurology, Tongji Hospital, Tongji Medical College, Huazhong University of Science and Technology, Wuhan 430030, P.R. China. [5]Faculty of Biology, Medicine and Health, The University of Manchester, Oxford Rd, Manchester M13 9PL, UK. [6]Achucarro Centre for Neuroscience, IKERBASQUE, Basque Foundation for Science, Plaza Euskadi 5, 48009 Bilbao, Spain. ✉e-mail: verena@sund.ku.dk; alexej.verkhratsky@manchester.ac.uk; nedergaard@sund.ku.dk

transmembrane $Cl^-$ gradient. Cytosolic $Cl^-$ concentration $[Cl^-]_i$ in neurones is low (~5 mM) but rises rapidly following bursts of GABAergic synaptic activity[21,22]. In the adult brain, astrocytes are believed to maintain, during rest, a high $[Cl^-]_i$ in the range of 20–40 mM[23–30], while $[Cl^-]_o$ is about 130 mM, thus setting the $Cl^-$ equilibrium potential ($E_{Cl}$) at ~–35 mV. The resting membrane potential of astrocytes is ~–80 mV and hence opening of astrocytic $GABA_A$ receptors ($GABA_AR$) generates $Cl^-$ efflux and depolarisation[25,31–33]. It has been suggested, but never proven, that efflux of $Cl^-$ from astrocytes may replenish $[Cl^-]_o$ at inhibitory synapses thereby supporting GABAergic transmission[34,35]. In support of this model, pharmacological blockade of astrocytic gap junctions in brain slices resulted in a faster decline of inhibitory transmission[36]. However, it remains unknown whether the ex vivo preparations accurately replicate astrocytic $[Cl^-]_i$ in vivo and how natural brain state in non-anesthetised mice controls astrocytic $[Cl^-]_i$. Here we demonstrate that astrocytic $[Cl^-]_i$ in awake behaving mice is high but changes dynamically along with neuronal activity. Optogenetic manipulations that increase or decrease astrocytic $[Cl^-]_i$ potentiate or supress sustained inhibitory transmission. Thus, astrocytes function as a reservoir of $Cl^-$ for GABAergic transmission; this astrocytic reservoir dynamically contributes to the inhibitory tone depending on brain states. These observations may have significant implication for disorders associated with failure of inhibitory transmission, such as seizure activity.

## Results

### Astrocytic $[Cl^-]_i$ changes rapidly in response to awakening

To monitor dynamic changes in astrocytic $[Cl^-]_i$ in awake freely moving or naturally sleeping mice, we used fibre photometry of astrocytes specifically expressing the fluorescent $Cl^-$ sensor mClY[37] under the *Gfap*-promoter to record the bulk signal from many astrocytes. mClY was delivered via AAV injection to the somatosensory cortex and an optic fibre was implanted above the injection site. In addition, electrodes were implanted for simultaneous EEG/EMG recordings (Fig. 1a). Two weeks after injection and surgery, the mClY signal was recorded early in the dark phase when the mice are mostly awake. The analysis showed that astrocytic $[Cl^-]_i$ depends on the state of the brain: $[Cl^-]_i$ exhibited dynamic fluctuations, with high frequency and irregular amplitudes during wakefulness, but stabilised during EEG/EMG-validated NREM sleep (Fig. 1b, c). On average $[Cl^-]_i$ was higher during sleep than during wakefulness (Fig. 1d). Transition from sleep to wakefulness was accompanied by a consistent decrease in $[Cl^-]_i$, conversely $[Cl^-]_i$ always increased during transition from wakefulness to sleep (Fig. 1e). Analysing multiple state transitions within individual mice confirmed that astrocytic $[Cl^-]_i$ directly depends on the brain state ($p = 0.0296$, Fig. 1f). Fluctuations in astrocytic $[Cl^-]_i$ signal, measured by standard deviation (SD) was significantly higher during wakefulness compared to sleep ($p = 0.0081$, Fig. 1f). Significantly higher astrocytic $[Cl^-]_i$ during sleep compared to wakefulness were demonstrated by comparing the distribution of all traces recorded from several mice ($p < 0.001$, Fig. 1g). As a negative control we recorded fluorescence of YFP expressed under the *Gfap*-promotor. In contrast to mClY, the average YFP signal did not differ between sleep and wakefulness and state transitions were not accompanied with YFP signal changes (Fig. 1h). Of note, mClY is based on YFP, but mClY is pH-independent in the physiological range in contrast to YFP[37] (Supplementary Figs. 1 and 10).

Additional analysis demonstrated significant correlation between astrocytic $[Cl^-]_i$ and EMG-based estimates of motor activity in the awake state. When the animal moves, EMG exhibits the highest SD, while in resting or asleep mice the SD is low. The $[Cl^-]_i$ decreased in parallel with an increase in EMG standard deviation ($p = 0.3$, Fig. 1j). Comparing the characteristic EEG power bands during sleep with astrocytic $[Cl^-]_i$ showed no correlation. In contrast, during wakefulness EEG theta power, which is associated with movement[32] is significantly positively correlated with astrocytic $[Cl^-]_i$ (Supplementary Fig. 2).

To analyse the impact of spontaneous locomotion on $[Cl^-]_i$, periods of mobile and immobile awake states were quantified by tracking videos of home cage mobility using EthoVision XT (Noldus). Overall, mobile and immobile periods were defined as follows: (i) intervals below 5% change in body area were detected as immobility, whereas (ii) intervals above 5% change in body area were considered as mobility periods[33]. Transitions that were preceded by a minimum of 10 s immobility and with mobile periods lasting at least 1 s were chosen for assessment of mClY fluctuations (Fig. 1k). A biphasic (short increase followed by a prolonged decrease) $[Cl^-]_i$ transient was consistently associated with the onset of locomotion (Fig. 1l).

Average astrocytic $[Cl^-]_i$ was higher during immobile versus mobile phases ($p = 0.0096$), with no changes in the SD of the signal (Fig. 1m). Comparing the distribution of all mClY traces recorded, confirmed that $[Cl^-]_i$ is higher during immobile phases ($p < 0.001$, Fig. 1n). The relative change of $[Cl^-]_i$ from sleep to wakefulness is larger compared to the relative change between rest and mobility (Fig. 1o).

### Onset of movement and sensory stimulation trigger a decrease in astrocytic $[Cl^-]_i$

We next tested whether astrocytic $[Cl^-]_i$ changes during locomotion. We recorded bulk astrocytic $[Cl^-]_i$ from the somatosensory cortex of head-fixed mice trained to run voluntarily on a Styrofoam sphere (Fig. 2a). Movements were tracked as front paw displacements while simultaneously recording $[Cl^-]_i$ (Fig. 2b). The $Cl^-$ signal was averaged over 4–28 movement onsets (Fig. 2b, c). Initiation of movement triggered a biphasic $[Cl^-]_i$ response comprising a short increase, followed by a slow, long-lasting decrease (Fig. 2d). Such $[Cl^-]_i$ dynamics are similar to those observed in freely running mice using fibre photometry (Fig. 1l). To test whether sensory stimulation also triggers changes in astrocytic $[Cl^-]_i$, trains of air puffs (30 s, 5 Hz, 50 ms pulse duration) were delivered to the whiskers while astrocytic $[Cl^-]_i$ was imaged in the contralateral barrel-field cortex (Fig. 2e). The $Cl^-$ signal was averaged over several stimulations (Fig. 2f, g). Whisker-induced astrocytic $[Cl^-]_i$ transients followed the same pattern as those triggered by movement. After a fast transient increase, $[Cl^-]_i$ drops and stays low for the duration of the stimulation (Fig. 2g).

Fluctuating $[Cl^-]_i$ during locomotion and in response to sensory stimulation suggests that astrocytic $[Cl^-]_i$ might be involved in cortical processing. Of particular interest is that both events trigger a similar response in astrocytic $[Cl^-]_i$ (Fig. 2h). The amplitude of initial increase, time to peak, negative dip amplitude, time to dip, slope of dip and the sum of positive and negative changes evoked by locomotion or whisker stimulation were all directly comparable (Fig. 2i).

### Optogenetic elevation of astrocytic $[Cl^-]_i$ suppresses neuronal activity

To investigate whether manipulations of astrocytic $[Cl^-]_i$ affect neuronal activity, the optogenetic $Cl^-$ pump Halorhodopsin (NpHR3.0) was expressed by injecting AAVs carrying the *Gfap*-NpHR3.0 construct to the somatosensory cortex (Fig. 3a). Neuronal activity was monitored by imaging bulk neuronal $[Ca^{2+}]_i$ using the redshifted $Ca^{2+}$ probe jRGECO1a[38], expressed by injecting the virus containing the construct *hSyn1*-jRGECO1a, in awake, head-fixed mice voluntarily running on a Styrofoam sphere. Constant stimulation of NpHR3.0 significantly increases astrocytic $[Cl^-]_i$ (Supplementary Figs. 3 and 4), while the stimulation light alone, applied to mice not expressing NpHR3.0, affected neither astrocytic $[Cl^-]_i$, nor brain activity (Supplementary Fig. 5). Onset of voluntary running in the presence of constant optogenetic stimulation of NpHR3.0 shows reduced astrocytic $Cl^-$ transients (Fig. 3b, c) indicative for increased $[Cl^-]_i$. Onset of voluntary running evoked an increase in neuronal $[Ca^{2+}]_i$ (Fig. 3d). Accumulation of $Cl^-$ into astrocytes by activation of NpHR3.0 modified neuronal $Ca^{2+}$ dynamics. Most notable the accelerated decay of the neuronal response when NpHR3.0 is activated. While the amplitude and time to

peak were not affected, the recovery time to baseline decreased significantly ($p = 0.0049$, Fig. 3e), and the decay of $[Ca^{2+}]_i$ transients accelerated ($p = 0.0384$, Fig. 3e). Neuronal $[Ca^{2+}]_i$ and astrocytic $[Cl^-]_i$ transients evoked by the onset of locomotion, show that the suppression of neuronal activity occurs concurrently with the peak of astrocytic $Cl^-$ efflux (Fig. 3c, e). Thus, optogenetic activation of

NpHR3.0 appears to counteract the decrease of astrocytic $[Cl^-]_i$ and thus maintains high $[Cl^-]_i$ during neuronal activity.

To further characterise the impact of astrocytic $[Cl^-]_i$ on neuronal activity, we manipulated astrocytic $[Cl^-]_i$ during whisker puff stimulation. This approach provided temporal control, which was lacking when studying spontaneous movement (Fig. 3b, d). Again,

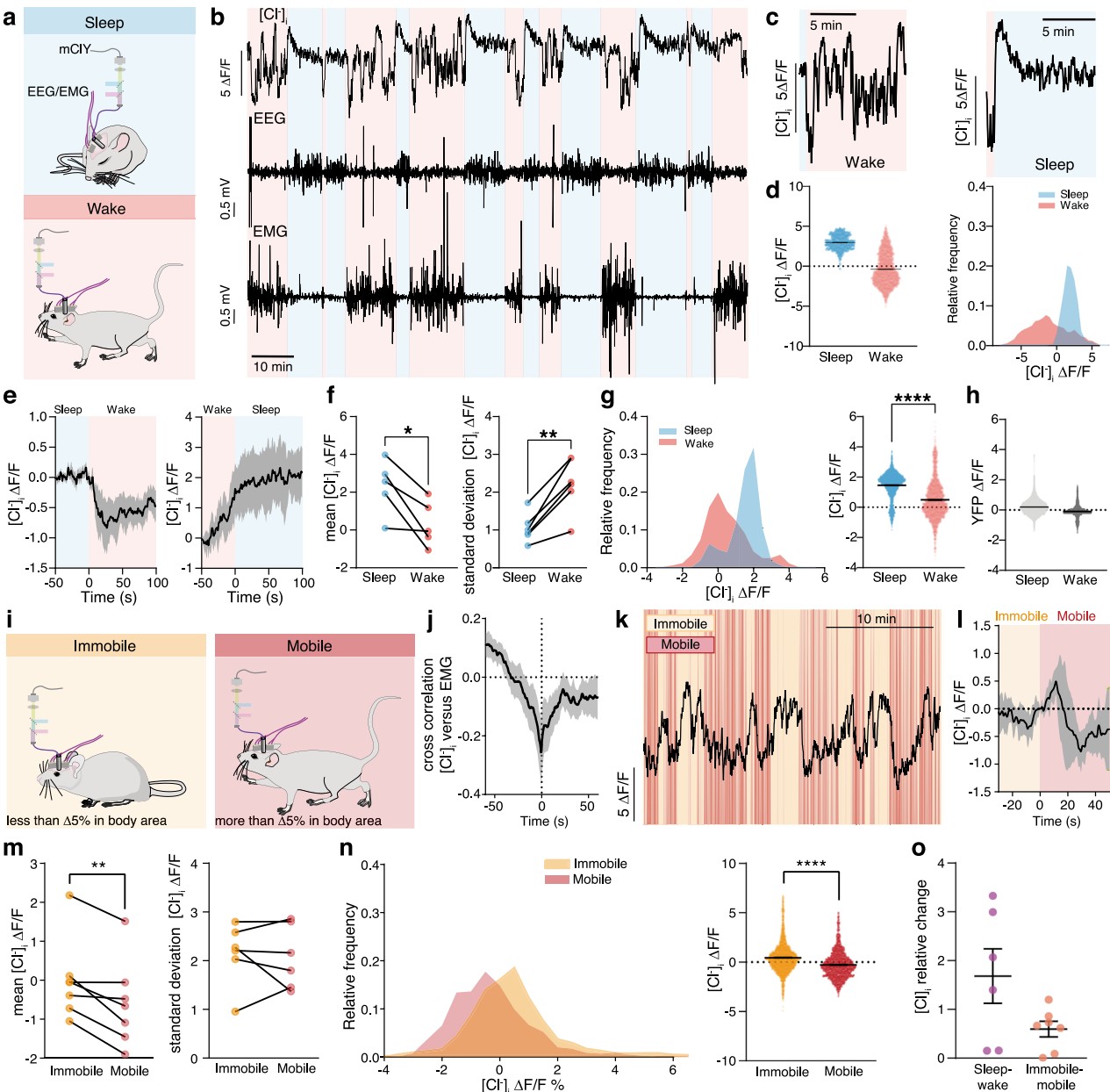

**Fig. 1 | Astrocytic $[Cl^-]_i$ is lower during wakefulness and fluctuates during locomotion. a** $[Cl^-]_i$ in cortical astrocytes was imaged using mClY and fibre photometry in combination with EEG/EMG recordings in awake, freely moving, or spontaneously sleeping mice. **b** Representative traces of astrocytic $[Cl^-]_i$, EEG, and EMG; colour code indicates sleep and awake periods. **c** Changes in $[Cl^-]_i$ during transiting from sleep to awake or from awake to sleep in expanded time scale. **d** Distribution of astrocytic $[Cl^-]_i$ during sleep and wakefulness. $N = 1$ representative mouse. **e** Average $[Cl^-]_i$ traces during transition from sleep to awake or awake to sleep, shading indicates ±SEM (standard error of the mean). $N = 6$ mice. **f** Mean $[Cl^-]_i$ and standard deviation (SD) during sleep and wakefulness. $N = 6$ mice, paired two-tailed $t$-test *$p = 0.0296$, **$p = 0.002$. **g** Distribution of $[Cl^-]_i$ in awake and sleep states recorded from freely moving and naturally sleeping mice. $N = 6$ mice, paired two-tailed $t$-test, ****$p < 0.001$. **h** Distribution of YFP recorded from freely moving and naturally sleeping mice. $N = 3$ mice. **i** $[Cl^-]_i$

in cortical astrocytes was imaged in awake and resting (immobile) or voluntary running (mobile, 10 s immobility followed by more than 1 s mobility) mice. **j** Cross correlation of $[Cl^-]_i$ versus SD of EMG. Data represent mean ± SEM. $N = 6$ mice, the average Pearson correlation coefficient: 0.258. **k** Representative traces of astrocytic $[Cl^-]_i$, EEG, and EMG; colour code indicates mobile and immobile periods. **l** $[Cl^-]_i$ trace during transition from immobile to mobile state, shading indicates ±SD. $N = 7$ mice. **m** Mean $[Cl^-]_i$ ($N = 7$ mice) and standard deviation ($N = 6$ mice) during immobile and mobile periods. Paired two-tailed $t$-test, **$p = 0.0096$, $p = 0.7490$. **n** Distribution of $[Cl^-]_i$ recorded from awake freely moving, mobile or immobile mice. $N = 6$ mice, one sample $t$-test, ****$p < 0.001$. **o** Relative changes of $[Cl^-]_i$ when transitioning between sleep and awake ($N = 6$ mice) versus immobile and mobile ($N = 7$ mice). Paired two-tailed $t$-test. $[Cl^-]_i = mClY - \Delta F/F$ (%). Data represent mean ± SEM. Source data are provided as a Source Data file.

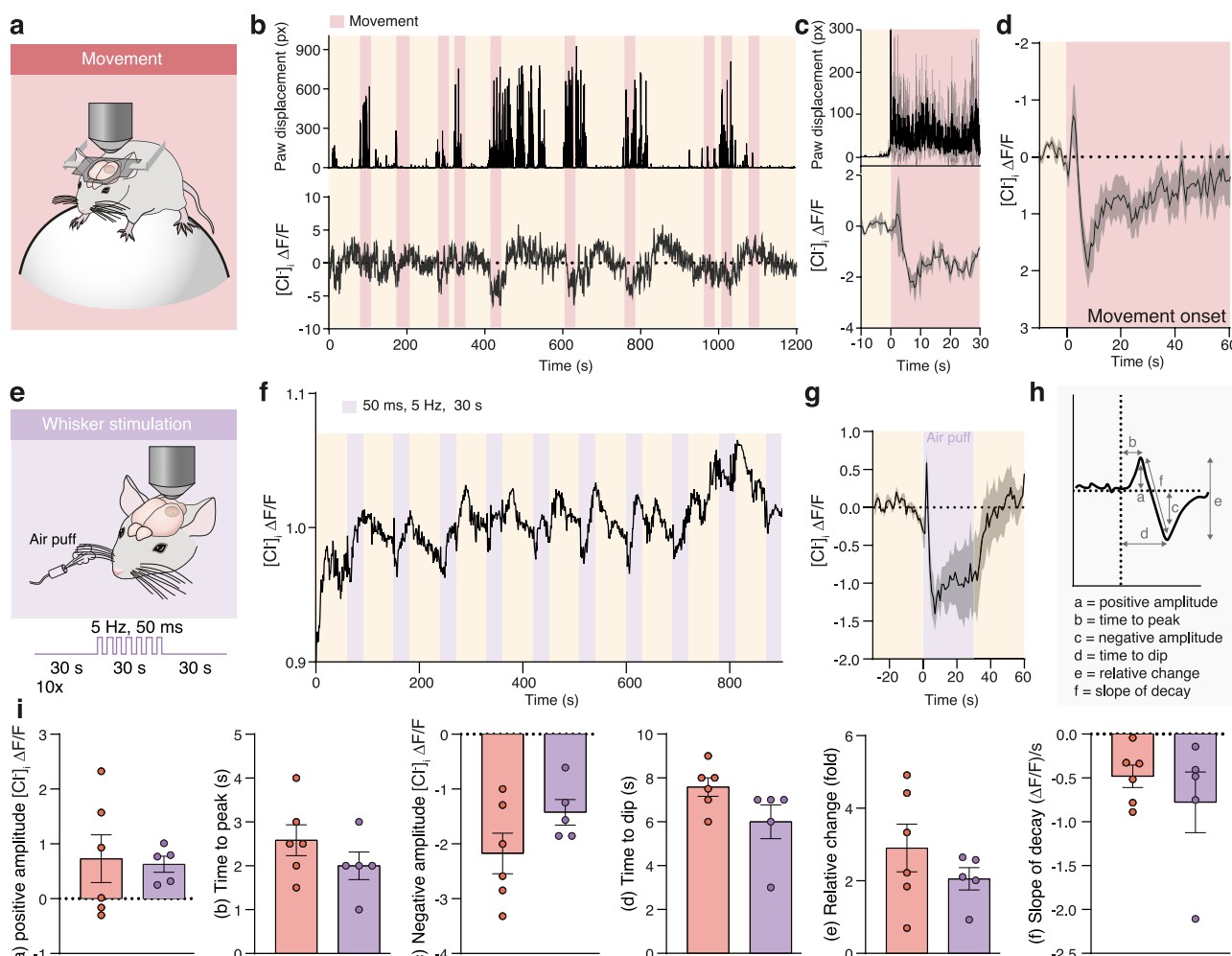

**Fig. 2 | Astrocytic [Cl⁻]ᵢ decreases upon movement and sensory stimulation.**
**a** Astrocytic [Cl⁻]ᵢ was imaged using mClY and macroscopic imaging from somatosensory cortex of awake head-fixed mice voluntarily running on a Styrofoam sphere. **b** Representative traces of front paw displacement (top) and astrocytic [Cl⁻]ᵢ (bottom). **c** Average of [Cl⁻]ᵢ traces (bottom) of **b** during multiple running periods aligned to the running onset (top), data represent mean ± SD. **d** Average [Cl⁻]ᵢ trace during movement onset, shading indicates ±SEM ($N$ = 6 mice). **e** Bulk astrocytic [Cl⁻]ᵢ was imaged in somatosensory cortex of awake stationary head-fixed mice subjected

to air-puff whisker stimulation. **f** Representative trace of astrocytic [Cl⁻]ᵢ upon whisker stimulation. **g** Average [Cl⁻]ᵢ trace during whisker stimulation, shading indicates ±SEM. $N$ = 4 mice. **h** Schematic of Cl⁻ signal analysis. **i** Comparison of the parameters of the stereotypical astrocytic [Cl⁻]ᵢ response to movement ($N$ = 6 mice) versus whisker stimulation ($N$ = 5 mice) depicted in panel **h**. Data represent mean ± SEM. Un-paired, two-tailed $t$-test, $p$ = 0.8341, $p$ = 0.2572, $p$ = 0.1387, $p$ = 0.0912, $p$ = 0.3032, $p$ = 0.4104. [Cl⁻]ᵢ = mClY − ΔF/F (%). Source data are provided as a Source Data file.

optogenetic stimulation was constantly employed to counteract the decrease of astrocytic [Cl⁻]ᵢ upon whisker stimulation (Fig. 3g). The air-puffing-induced neuronal [Ca²⁺]ᵢ increase lasted for the length of the stimulation (Fig. 3i). In the presence of constantly stimulated NpHR3.0, air-puffing onto the whiskers triggered significantly reduced astrocytic Cl⁻ transients (Fig. 3g, h). Accumulation of Cl⁻ into astrocytes by activation of NpHR3.0 modified neuronal Ca²⁺ dynamics. Most notable is the suppressed plateau phase of the neuronal response when NpHR3.0 is activated. While the amplitude and time to peak were not affected, the amplitude of the plateau decreased significantly (Fig. 3j). Astrocytic [Cl⁻]ᵢ and neuronal [Ca²⁺]ᵢ transients show that the largest change occurred 6–8 s after the beginning of whisker stimulation (Fig. 3h, j). A significant reduction in neuronal [Ca²⁺]ᵢ occurred concomitantly with the dip in astrocytic [Cl⁻]ᵢ (Fig. 3h, j). Light-induced increase in astrocytic [Cl⁻]ᵢ consistently decreased the neuronal response to whisker stimulation (Fig. 3j).

We conclude that manipulations which increase astrocytic [Cl⁻]ᵢ shorten neuronal activity. While the initial timing and the maximal amplitude of neuronal [Ca²⁺]ᵢ are not changed, elevations of

astrocytic [Cl⁻]ᵢ have a significant impact on the plateau of the neuronal response and accelerate the return of neuronal [Ca²⁺]ᵢ to the baseline following both spontaneous locomotion and whisker stimulation. Incidentally, allosteric GABA_AR agonist Diazepam affects neuronal [Ca²⁺]ᵢ transient similarly to the activation of NpHR3.0 in astrocytes (Supplementary Fig. 6). Furthermore, inhibitory postsynaptic current (IPSC) recorded in acute brain slices upon electrical stimulation show an increase when NpHR3.0 in astrocytes is activated (Supplementary Fig. 7). These findings support the notion that astrocytic [Cl⁻]ᵢ serves as a source of Cl⁻ for GABAergic transmission, since transient increases in excitatory transmission are terminated faster when astrocytic [Cl⁻]ᵢ is high in awake behaving mice.

### Optogenetic depletion of astrocytic [Cl⁻]ᵢ facilitates excitatory neuronal activity
To investigate whether decreased baseline astrocytic [Cl⁻]ᵢ affects neuronal activity, the optogenetic switchable Cl⁻ channel SwiChR++[39] was expressed in astrocytes by injecting AAVs carrying the *Gfap*-SwiChR++ construct to somatosensory cortex (Fig. 4a). SwiChR++ is opened by cyan (470 nm) light and closed by red (600 nm) light. Using

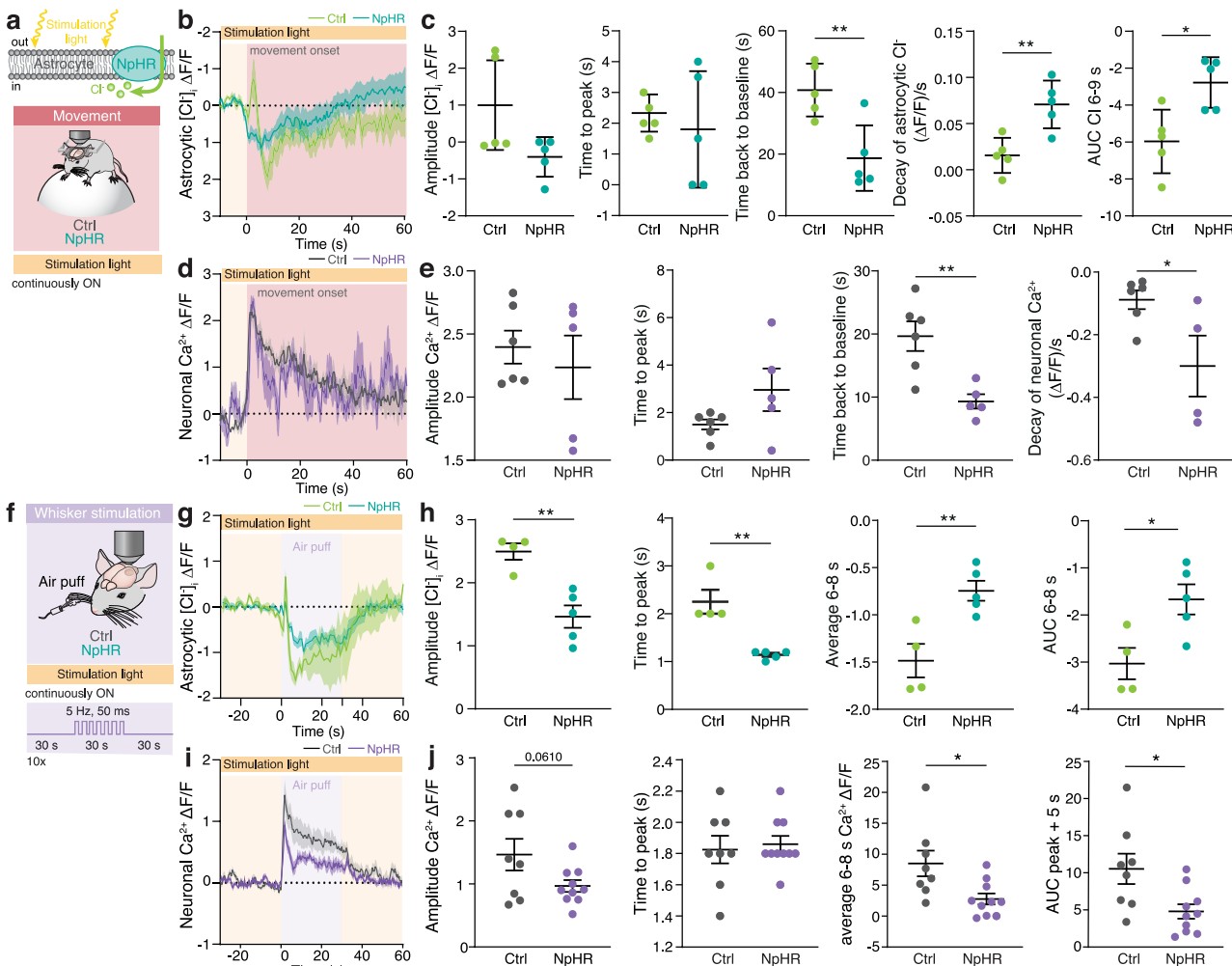

**Fig. 3 | Optogenetic elevation of astrocytic [Cl⁻]ᵢ shortens activation-induced neuronal [Ca²⁺]ᵢ. a** The optogenetic Cl⁻ pump was expressed in astrocytes to manipulate [Cl⁻]ᵢ, while astrocytic [Cl⁻]ᵢ or neuronal [Ca²⁺]ᵢ was imaged. As a negative control, light stimulation was applied to mice not expressing NpHR3.0. Mice were voluntarily running on a Styrofoam sphere. **b** Average astrocytic [Cl⁻]ᵢ traces during transition from stationary to mobile, recorded during continuous light stimulation of NpHR3.0 in astrocytes with or without NpHR3.0; shading indicates ±SEM. N = 6 ctrl/5 NpHR3.0. **c** Peak amplitude of astrocytic [Cl⁻]ᵢ upon movement onset; time to peak of astrocytic [Cl⁻]ᵢ; time to baseline; slope of decay, and area under the curve (AUC) from 6 to 8 s, data represent mean ± SEM. N = 5 ctrl/5 NpHR3.0, un-paired two-tailed t-test. p = 0.9238, p = 0.6626, **p = 0.0068, **p = 0.0049 *p = 0.0119. **d** Average neuronal [Ca²⁺]ᵢ trace during transition from stationary to mobile, recorded during continuous light stimulation of NpHR3.0 in astrocytes with or without NpHR3.0; shading indicates ±SEM. N = 3 mice. **e** Peak amplitude of neuronal [Ca²⁺]ᵢ upon movement onset; time to peak; recovery time to baseline; slope of decay, data represent mean ± SEM. N = 6 ctrl/5 NpHR3.0,

un-paired two-tailed t-test, p = 0.5665, p = 0.1167, **p = 0.0049, *p = 0.0384. **f** Using the same protocol as in Fig. 2, whiskers were stimulated using air puffs. **g** astrocytic [Cl⁻]ᵢ trace during whisker stimulation, recorded during continuous light stimulation of NpHR3.0 in astrocytes with or without NpHR3.0; shading indicates ±SEM. N = 4 ctrl/5 NpHR3.0. **h** Peak amplitude of astrocytic [Cl⁻]ᵢ upon whisker stimulation; time to peak; period of maximal changes, 6–8 s after onset of stimulation. AUC during 6–8 s after onset of stimulation, data represent mean ± SEM. N = 4 ctrl/ 5 NpHR3.0, un-paired two-tailed t-test, **p = 0.0030, **p = 0.0017, **p = 0.0071, *p = 0.022. **i** Average neuronal [Ca²⁺]ᵢ trace during whisker stimulation, recorded during continuous light stimulation of NpHR3.0 in astrocytes with or without NpHR3.0; shading indicates ±SEM. N = 8 ctrl/10 NpHR3.0. **j** Peak amplitude of neuronal [Ca²⁺]ᵢ upon whisker stimulation; time to peak; period of maximal astrocytic [Cl⁻]ᵢ changes, 6–8 s after onset of stimulation. AUC of neuronal [Ca²⁺]ᵢ during 5 s after peak, data represent mean ± SEM. N = 8 ctrl/10 NpHR3.0, un-paired two-tailed t-test, p = 0.061, p = 0.7249, *p = 0.0147, *p = 0.0155. [Cl⁻]ᵢ = mCIY − ΔF/F (%). Source data are provided as a Source Data file.

two-photon imaging, astrocytic [Cl⁻]ᵢ in individual cells was monitored using mCIY (Fig. 4b). Neuronal activity was monitored by imaging single cell neuronal [Ca²⁺]ᵢ using GCaMP6s expressed under the *Thy1* promotor of transgenic animals (Fig. 4d, e).

To characterise the impact of reduced astrocytic [Cl⁻]ᵢ on neuronal activity, we manipulated astrocytic [Cl⁻]ᵢ during whisker puff stimulation. Optogenetic activation of SwiChR++ reduced astrocytic [Cl⁻]ᵢ response to whisker stimulation (Fig. 4c); this effect is most prominent during the late phase of whisker stimulation. Activation of SwiChR++ in astrocytes increased the plateau phase of neuronal [Ca²⁺]ᵢ transients (Fig. 4e). Neither the amplitude nor the time to peak of neuronal [Ca²⁺]ᵢ responses were affected by activation of

SwiChR++ in astrocytes. A significant increase in neuronal [Ca²⁺]ᵢ occurred concomitantly with the disappearance of the dip in astrocytic [Cl⁻]ᵢ during the late phase of whisker stimulation (Fig. 4f).

We conclude that in vivo manipulations which decrease astrocytic [Cl⁻]ᵢ enhance excitatory activity. Using IPSC recordings in acute brain slices in response to electrical stimulation we found that SwiChR++ activation and thereby decreasing [Cl⁻]ᵢ in astrocytes reduced IPSC amplitude (Supplementary Fig. 7). These findings support the notion that astrocytic [Cl⁻]ᵢ serves as a source of Cl⁻ for GABAergic transmission. Transient increases in excitatory transmission in awake behaving mice are prolonged when astrocytic [Cl⁻]ᵢ is low.

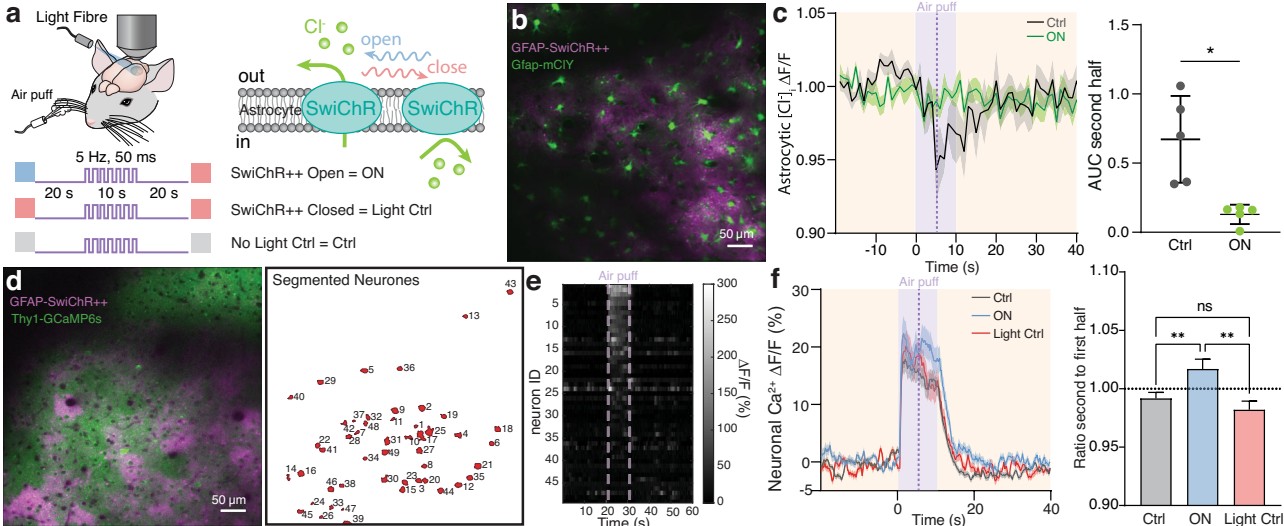

**Fig. 4 | Optogenetic depletion of astrocytic [Cl⁻]ᵢ extends activation-induced neuronal [Ca²⁺]ᵢ signal. a** The optogenetic switchable Cl⁻ channel was expressed in astrocytes to manipulate [Cl⁻]ᵢ, while astrocytic [Cl⁻]ᵢ or neuronal [Ca²⁺]ᵢ were imaged using mClY or GCaMP6s, respectively. As a negative control, no light or only red-light stimulation was applied to mice expressing SwiChR++. The awake, head-fixed mice were whisker stimulated using air puffs. **b** Average projection of Gfap-mClY and Gfap-SwiChR++ imaged on the two-photon microscope (2PM). **c** Average astrocytic [Cl⁻]ᵢ trace during whisker stimulation in control, light control and upon light-activation of SwiChR++ in astrocytes, shading indicates ±SD. AUC during the second half (last 5 s) of whisker stimulation, error bars indicate ±SD. $N = 5$ mice, paired two-tailed $t$-test, *$p = 0.0278$. **d** Average

projection of Thy1-GCamp and Gfap-SwiChR++ on the 2PM and ROIs of single cells that were detected. **e** Raster plot showing the activation profile of the neurones detected in **d**, recordings were repeated 5 times per animal. **f** Average neuronal [Ca²⁺]ᵢ trace during whisker stimulation in control, light control and upon light-activation of SwiChR++ in astrocytes, shading indicates ±SEM. Ratio between mean signal of the first (5 s) and second half (last 5 s) of neuronal [Ca²⁺]ᵢ. Data represent mean ± SEM. $N = 6$ mice, 560 neurones, one-way ANOVA with a Geisser-Greenhouse correction Ctrl versus ON **$p = 0.0099$, Ctrl versus OFF $p = 0.4106$, ON versus OFF **$p = 0.0015$. [Cl⁻]ᵢ = mClY − ΔF/F (%). Source data are provided as a Source Data file.

## Activation of GABA_A receptors triggers Cl⁻ efflux from astrocytes

We tested the hypothesis that activation of GABA_AR during locomotion and whisker stimulation is responsible for efflux of astrocytic [Cl⁻]ᵢ[34–36]. Astrocytes in several brain regions including cortex express GABA_ARs[25,35,40–42] (Supplementary Table 1), which are often localised around inhibitory synapses[43] raising the possibility that GABA activates astrocytic GABA_ARs close to the synaptic cleft thereby supplying Cl⁻ to sustain inhibitory transmission. To activate GABA_ARs, either GABA or the GABA_AR agonist muscimol were applied by pressure injection through a micropipette (500 μM, 1–3 μm tip diameter, 2–8 psi, 100 ms) as described previously[44], while recording fluorescence intensity of mClY or MQAE (Fig. 5a). Opening of GABA-gated anion channel leads to Cl⁻ movement along the diffusion gradient, hence changes in [Cl⁻]ᵢ reveal the direction of Cl⁻ fluxes.

The genetically encoded Cl⁻ sensor mClY was expressed by injection of AAVs carrying the *Gfap*-mClY construct into somatosensory cortex (Fig. 4b). Muscimol induced a transient decrease in [Cl⁻]ᵢ in awake animals (Fig. 5c, d) that lasted for 8.36 ± 3.1 s (Fig. 5e).

This observation was next tested using an alternative approach: Astrocytes loaded with the Cl⁻ dye MQAE were imaged while cell type identity was confirmed by co-labelling with astrocyte-specific dye SR-101[45,46] which allowed to focus on astrocytes to exclude contamination from neuronal signalling (Fig. 5f). MQAE is insensitive to bicarbonate and pH changes, while mClY is Cl⁻ selective at physiological pH ranges[37]. Fluorescence intensity as well as fluorescence lifetime directly report the [Cl⁻]ᵢ. Absolute [Cl⁻]ᵢ imaging of fluorescence lifetimes in awake mice in vivo shows a normal distribution as reported previously for brain slices and cell culture[29,30] (Supplementary Fig. 3). Imaging of MQAE fluorescent intensity revealed that GABA administration reduced [Cl⁻]ᵢ corroborating the mClY data (Fig. 5g). The transient decrease of [Cl⁻]ᵢ was rapid and lasted for 15.10 ± 10.1 s (Fig. 4h, i). Muscimol also triggers transient decrease in [Cl⁻]ᵢ with an

average duration of 14.89 ± 7.6 s ($N = 10$ mice, Fig. 4h, i). As a negative control, aCSF was pressure injected through a micropipette with the same settings used for GABA and muscimol administration. In contrast to muscimol and GABA injections, aCSF produces a fast artefact lasting 1.2 ± 0.73 s and does not affect [Cl⁻]ᵢ (Fig. 4j, k, l). These observations show that basal astrocytic [Cl⁻]ᵢ in vivo is above the diffusion equilibrium of 7 mM Cl⁻ by which Cl⁻ channel opening triggers Cl⁻ efflux.

We can conclude that astrocytes respond to GABA by mounting Cl⁻ efflux. Imaging with two different Cl⁻ indicators, the Cl⁻ sensor mClY and the Cl⁻-sensitive dye MQAE, leads to the same conclusion. The Cl⁻ efflux might be directed into the synaptic cleft based on the relative higher density of GABA_AR around inhibitory synapses than elsewhere[43]. In this way, astrocytes provide a dynamic reservoir for Cl⁻ that can be recruited by activation of astrocytic GABA_AR or other anion channels. Thus, astrocytes can broadly boost inhibitory transmission on a slower time scale than synaptic transmission itself.

## Discussion

In this study, we found that astrocytes serve as a source of Cl⁻ required for sustained GABAergic transmission. We provided in vivo analysis of astrocytic [Cl⁻]ᵢ using four imaging approaches including two-photon microscopy (2PM), fluorescence lifetime imaging microscopy (FLIM), fibre photometry, and macroscopic imaging. We showed that astrocytic [Cl⁻]ᵢ in vivo is above the diffusion equilibrium (Fig. 5), and astrocytic [Cl⁻]ᵢ is dependent on the brain state with high [Cl⁻]ᵢ during natural sleep and lower, but highly dynamic [Cl⁻]ᵢ during wakefulness (Fig. 1). A key observation is that astrocytic [Cl⁻]ᵢ rapidly declines during neuronal firing induced by either spontaneous locomotion or whisker stimulation (Fig. 2). Perhaps even more importantly, the analysis documented that those changes in astrocytic [Cl⁻]ᵢ modulate neuronal activity (Figs. 3 and 4). Protoplasmic astrocytes in different brain regions, including cortex, are known to express GABA_AR[40–43,47–50].

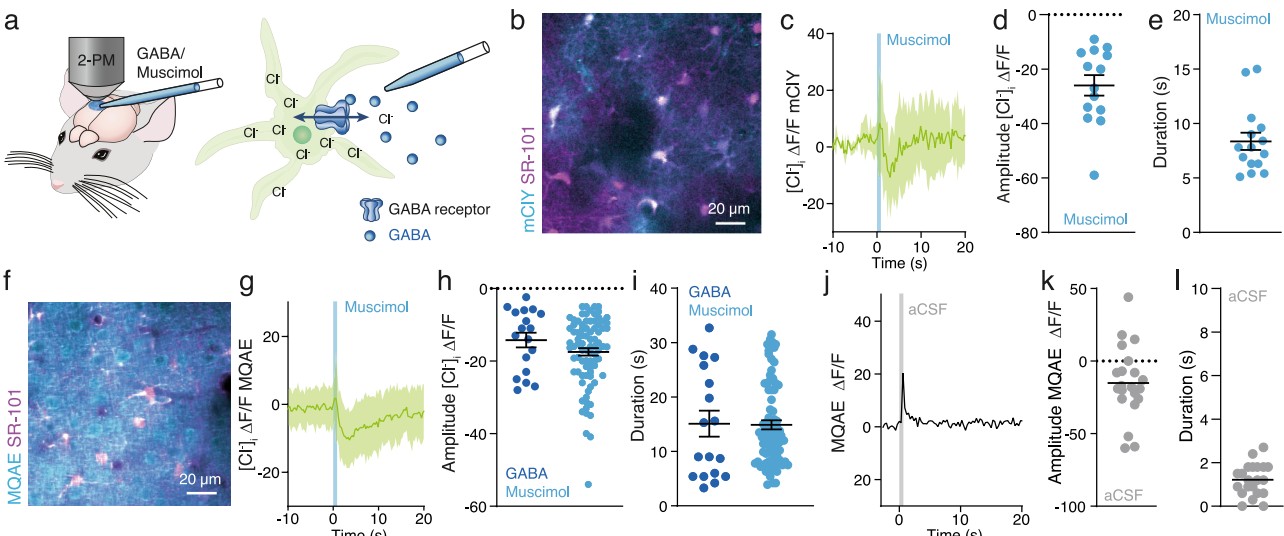

**Fig. 5 | Activation of GABA_A receptors in awake mice, trigger Cl⁻ efflux from astrocytes. a** Astrocytic [Cl⁻]_i recordings using mClY or MQAE in response to injections of GABA or muscimol. **b, f** Representative images of the mouse cortex expressing mClY in astrocytes or loaded with MQAE and co-labelling with SR-101. **c, g** Astrocytic [Cl⁻]_i traces during muscimol injection, shading indicates ±SD. *N* = 3 mice. **d, h** Decrease of astrocytic [Cl⁻]_i in response to muscimol or GABA injections. Data represent mean ± SD. *N* = 4 mClY/Muscimol, 3 mice MQAE/GABA, 10 mice MQAE/Muscimol. **e, i** Duration of astrocytic [Cl⁻]_i transient upon GABA or muscimol injections. Data represent mean ± SD. *N* = 4 mClY/Muscimol, 3 mice MQAE/GABA, 10 mice MQAE/Muscimol. **j** Astrocytic [Cl⁻]_i trace during aCSF injection, mean ± SD. *N* = 1 mouse, 21 cells. **k** Change in astrocytic [Cl⁻]_i in response to aCSF injection. *N* = 1 mouse, 21 cells. **l** Duration of astrocytic [Cl⁻]_i change upon aCSF injection. *N* = 1 mouse, 21 cells. [Cl⁻] = mClY − ΔF/F (%). Data represent mean ± SD. Source data are provided as a Source Data file.

It is conceivable to hypothesise that activation of astrocytic GABA_ARs generates Cl⁻ efflux that increases [Cl⁻]_o thus strengthening inhibitory transmission during prolonged episodes of neuronal activity. Optogenetic manipulation which increases baseline astrocytic [Cl⁻]_i results in reduced neuronal activity (Fig. 3), while depleting astrocytic [Cl⁻]_i results in enhanced neuronal activity (Fig. 4). Electrophysiological recordings confirm that manipulation with astrocytic [Cl⁻]_i directly affects IPSCs (Supplementary Fig. 7). Combined, these observations document that astrocytes modulate neuronal transmission within a slow time-frame of seconds. Our observations provide additional support to the notion that astrocytes are more tightly linked to GABAergic inhibition, than to the glutamatergic excitatory transmission in the developing and mature brain[51–55].

It is well known that long-lasting activation of inhibitory synapses leads to a significant reduction in the amplitude of IPSPs over time, reflecting a collapse of the Cl⁻ gradient due to Cl⁻ influx into neurones through GABA_AR[21,56,57]. Voltage-clamp recordings from neocortical pyramidal cells and dendrites of hippocampal CA1 pyramidal cells revealed a shift in the reversal potential of GABA_AR currents during sustained inhibition indicating a decrease in transmembrane Cl⁻ gradient[21,56]. Whole cell patch-clamp recordings, in combination with [Cl⁻]_i imaging of somata of mature hippocampal CA1 pyramidal neurones, confirmed synaptically activated GABA_AR-mediated cytoplasmic Cl⁻ accumulation. Decrease in the neuronal Cl⁻ gradient is accelerated by pharmacological inhibition of astrocytic gap junctions in ex vivo brain slices[36]. Furthermore, recordings with Cl⁻ sensitive microelectrodes showed that stimulation of CA3 pyramidal neurones leads to an increase in [Cl⁻]_o, which is reduced by applying a gap junction blocker[58]. We here show that manipulating astrocytic [Cl⁻]_i by optogenetic stimulation of the Cl⁻ pump reduced neuronal activity during locomotion or whisker stimulation resulting in a decrease of plateau and faster recovery of neuronal [Ca²⁺]_i transients. A similar potentiation of inhibitory transmission could be evoked by pharmacologically activating GABA_AR using the allosteric modulator of GABA_AR Diazepam (Supplementary Fig. 6). Diazepam had a comparable effect on neuronal activity as optogenetic manipulation of astrocytic [Cl⁻]_i supporting

the notion that astrocytic [Cl⁻]_i modulates inhibition by a GABA_AR mediated mechanism. Accordingly, IPSC recordings upon electrical stimulation show a slower decrease when baseline [Cl⁻]_i is increased, indicative of reduced inhibitory activity in acute brain slices (Supplementary Fig. 7). To confirm a GABA_AR mediated mechanism future studies may benefit from astrocyte-specific manipulation of GABA_AR by either silencing or a conditional knock-out mutant. However, such effort is complicated since astrocytes express 19 GABA_AR subunits (Supplementary Table 1).

Upon optogenetic stimulation Cl⁻ is constantly pumped into astrocytes expressing NpHR3.0 thus maintaining high astrocytic [Cl⁻]_i during movement or whisker stimulation. When astrocytic [Cl⁻]_i is elevated by optogenetic stimulation, the GABA-mediated Cl⁻ efflux from astrocytes is increased. While NpHR3.0 is expressed throughout the astrocytic membrane, as shown by 2PM imaging, astrocytic GABA_ARs are predominantly localised around the synaptic cleft[36,43]. Thus, Cl⁻ efflux is likely to occur with a high spatial precision in the vicinity of inhibitory synapses[36,43]. Electrophysiological LFP recordings during manipulations of [Cl⁻]_o show significant increases in baseline activity, demonstrating the powerful role of [Cl⁻]_o in modulating neuronal activity (Supplementary Fig. 5). Conversely, depleting astrocytic Cl⁻ gradient using a light switchable Cl⁻ channel boosted excitatory neuronal activity during whisker stimulation resulting in a prolonged plateau of neuronal [Ca²⁺]_i transients (Fig. 4). Furthermore, IPSCs in acute brain slices exhibited faster decrease when astrocytes were unable to provide Cl⁻, indicative of reduced inhibition upon electrical stimulation (Supplementary Fig. 7). It is unlikely that optogenetic elevation of astrocytic [Cl⁻]_i altered neuronal membrane potential since the photo-stimulation was not associated with changes in LFP (Supplementary Fig. 5). Extracellular [Cl⁻] is high and ranges around 130 mM. Stimulation of the optogenetic Cl⁻ pump homogeneously distributed in astrocytes, mediates global uptake of Cl⁻ from the extracellular space, which arguably has little impact on bulk [Cl⁻]_o. Furthermore, Cl⁻ crosses the blood-brain barrier with ease[59]. In quiet wakefulness, activation of NpHR3.0 in astrocytes did not modify spontaneous neuronal activity, suggesting that during baseline conditions, the level of tonic inhibition is not affected (Supplementary

**Fig. 6 | Astrocytic [Cl⁻]ᵢ facilitates neuronal inhibition by supplying Cl⁻ for GABAergic synapses: the hypothesis.** Synaptic GABA release opens GABA_AR in astrocytes leading to Cl⁻ efflux. Astrocytic [Cl⁻]ᵢ affects the duration of neural activation. Under conditions of low astrocytic [Cl⁻]ᵢ, neuronal activity is sustained for a longer period of time due to a reduction in the strength of GABAergic transmission over time. When astrocytic [Cl⁻]ᵢ is high, activation of astrocytic GABA_ARs provides an additional source of Cl⁻ shortening the duration of neuronal activity. The arrow indicates onset of cortical neural activity induced by spontaneous locomotion or whisker stimulation. Of note, Cl⁻ efflux from astrocytes may potentially be mediated by other molecular pathways including several types of anionic channels.

Fig. 5). During periods of neuronal activity, GABA released from inhibitory terminals opens astrocytic GABA_AR[36], thus triggering local Cl⁻ release into the synaptic cleft (Fig. 6). Even though we cannot provide proof for the mechanism, our report provides direct evidence that astrocytic [Cl⁻]ᵢ modulates neuronal activity in vivo (Figs. 3 and 4) and that manipulating astrocytic [Cl⁻]ᵢ consistently affects neuronal firing during locomotion or whisker stimulation through modulating inhibitory transmission in awake behaving mice (Fig. 5).

Previous studies estimated astrocytic [Cl⁻]ᵢ to range between 29 and 51 mM, depending on brain region, age and experimental setup[23–30]. Yet, hitherto astrocytic [Cl⁻]ᵢ in awake behaving mice was not measured. We here report that in vivo, astrocytic [Cl⁻]ᵢ is highly dynamic and depends on the brain state. A stereotypical pattern of dynamic changes in astrocytic [Cl⁻]ᵢ is observed upon arousal and locomotion when cortical networks are desynchronised. Rapid increase in astrocytic [Cl⁻]ᵢ is followed by a prolonged decrease which terminates with the respective stimulus. Furthermore, astrocytic [Cl⁻]ᵢ is increasing before transitioning to sleep and remains high during NREM sleep. Previous studies described neural activity in cortex to be lower during sleep compared to wakefulness[59]. Thus, the demand for astrocytic Cl⁻ release is low, possibly contributing to an increase in astrocytic [Cl⁻]ᵢ, during sleep. Future studies must address whether the changes in astrocytic Cl⁻ are causally involved in state-dependent changes in neural activity or merely result from changes in neural activity patterns.

Triggering neuronal activity by whisker stimulation or movement onset consistently evoked a biphasic response of astrocytic [Cl⁻]ᵢ (Fig. 2). Fast increase in astrocytic [Cl⁻]ᵢ could be explained by Cl⁻ influx through, for example, the glycine transporter GlyT1, which is reported to have the same time course as K⁺ uptake through K2P channels[60]. The transporter NKCC1 can potentially be involved too[61], yet NKCC1 is not expressed in astrocytes in the adult forebrain[62,63]. Volume changes mediated by increases in extracellular K⁺ are unlikely to trigger a transient increase in [Cl⁻]ᵢ, as astrocytic swelling upon physiological increases of K⁺ develops within much slower time range of minutes[63]. We hypothesise that a prolonged decrease of astrocytic [Cl⁻]ᵢ reflects opening of GABA_AR, since puff application of GABA or muscimol triggers a similar Cl⁻ release from astrocytes (Fig. 5). There are however other possible molecular pathways which may mediate Cl efflux from astrocytes, including opening of anion channels such as bestrophin-1 (Best-1)[60] or pannexin-1 (Panx1)[64]. We may note, however, that electrical stimulation of the Schaffer collateral pathway results in the prolonged decrease in astrocytic [Cl⁻]ᵢ mediated by Best-1 Cl⁻ channel in brain slices[60], but the observed slow decay (70–80 s) is several fold longer than the 10–15 s observed in our experiments (Fig. 2). Evoked astrocytic Ca²⁺ transients start a few seconds after stimulus onset and last much longer than the stimulation. In comparison, astrocytic Cl⁻

transients observed by us start immediately upon initiation of the stimulation, and terminate quickly after the stimulation stops. This may suggest that astrocytic Cl⁻ and Ca²⁺ transients are not directly correlated. Additional experiments with specific silencing of various channels are needed to fully resolve this question.

In summary, we demonstrated that astrocytic Cl⁻ modulates neuronal signalling in vivo. Our observations highlight the importance of astrocytic Cl⁻ for regulation of the excitation-inhibition balance. We propose that astrocytes serve as a reservoir of Cl⁻ that is recruited by activation of GABA_AR and act as slow modulator of neuronal signalling by replenishing Cl⁻ during periods of prolonged neuronal activity.

Excitatory actions of GABA were reported in about 30% of neurones from brain slices of patients with epilepsy[65] as well as in vitro using a variety of convulsive agents and procedures[66–68]. An increase of neuronal [Cl⁻]ᵢ occurs after seizures, spinal cord lesions, and other pathological conditions. Most studies investigating excitatory effects of GABA address neuronal [Cl⁻]ᵢ and related regulatory mechanisms to NKCC1 and KCC2[69]. Our study highlights contribution of astrocytic [Cl⁻]ᵢ and astrocytic Cl⁻ conductance. Dynamic changes of [Cl⁻] in the synaptic cleft define the activity-dependent disinhibition, which can vary from one neuronal compartment to another or be globally regulated by astrocytes. Our data suggests that long-lasting inhibition depends on astrocytic GABA_AR and astrocytic [Cl⁻]ᵢ (Fig. 6). A decrease in either is expected to lower the threshold for seizure induction.

## Methods
### Animals and surgery
All experiments conducted at University of Copenhagen were approved by the Danish Animal Experiments Inspectorate and were overseen by the University of Copenhagen Institutional Animal Care and Use Committee (IACUC), in compliance with the European Communities Council Directive of 22 September 2010 (2010/63/EU) legislation governing the protection of animals used for scientific purposes. License number 2016-15-0201-01030, 2021-12-0201-01038, and 2020-15-0201-00562. All experiments conducted at the University of Rochester were approved by the University Committee on Animal Resources of the University of Rochester Medical Centre (Protocol No. 2011-023) and an effort was made to minimise the number of animals used. This study used C57/BL6 WT mice, purchased from Janvier or Charles River. Thy1-GCaMP mice were purchased from Jackson laboratories (strain 024275) and bred in-house. Surgery was performed on mice 7–10 weeks old anaesthetised with Isoflurane (26675-46-7) (4% induction, 1.5% maintenance) or Ketamine (6740-88-7)/Xylazine (7361-61-7) (10 mg/ml and 1 mg/ml, respectively in 0.9% saline, 0.1 ml/mg bodyweight i.p.). Body temperature was monitored and maintained at 37 °C.

**Table 1 | The list of viruses used in this study**

| Construct | Company |
|---|---|
| ssAAV-PHP.eB/2-GFAP(2.2)-mClY-WPRE-bGHp(A) | Viral Vector Facility (VVF), Neuroscience Centre Zurich |
| pAAV-GFAP-mClY-N1-WPRE | Addgene plasmid #90457 |
| ssAAV-PHP.eB/2-hGFAP(0.7)-eNpHR3.0_EGFP-WRPE-bGHp(A) | Viral Vector Facility (VVF), Neuroscience Centre Zurich |
| pAAV-GFAP(0.7)-eNpHR3.0-eGFP | Vector Biolabs plasmid #VB2451 |
| PHP.eB-pAAV-mGFAP(ABC1D-eNpHR3.0-mCherry wPRE | Charité Viral Core Facility |
| ssAAV-PHP.eB/2-hGFAP-hHBbl/E-SwiChR++_TS_mTagBFP2-WPRE-bGHp(A) | Viral Vector Facility (VVF), Neuroscience Centre Zurich |
| AAV1-Syn-FLEX-jRGECO1a-WPRE-SV40 | A gift from Douglas Kim & GENIE Project (Addgene viral prep #100852-AAV1) |
| pAAV.CAG.Flex.NES-jRGECO1a.WPRE.SV40 | A gift from Douglas Kim & GENIE Project (Addgene plasmid #1000852) |
| AAV5-GFAP-eGFP | a gift from James M. Wilson (Addgene viral prep # 105549-AAV5) |
| pAAV5-GFAP-eGFP | a gift from James M. Wilson (Addgene plasmid # 105549) |
| ssAAV-PHP.S/2-shortCAG-dlox-EYFP(rev)-dlox-WPRE-hGHp(A) | Viral Vector Facility (VVF), Neuroscience Centre Zurich |
| AAV9-GFAP-Cre | Viral Vector Core, Gunma University Initiative for Advanced Research (GIAR) GA-030 |

## Fibre implantation

Before fibre implantation, virus injection was performed using a 10 μL glass Hamilton syringe (NF35BV-2, Nanofil, WPI), mounted to a Nanoinjector pump (Micro4, WPI). At stereotaxic coordinates A/P: −2.53 mm, M/L: −1.5, V/D: −0.7, −0.6, −0.5 mm from bregma a concentration of $2.5*10^{11}$–$1*10^{13}$ GC/ml was injected. A glass fibre was implanted in the centre of the injection site. Two EEG electrodes were implanted via two burr wholes and two EMG wires were implanted into the neck muscle. Animals were imaged 2–4 weeks after injection.

## Stereotaxic AAV injections

Virus injection was performed with the Hamilton syringe mounted to a micromanipulator (World Precision Instruments) at a 10-degree angle. At the coordinates A/P: −2.98 mm, M/L: −3.00 mm from bregma a concentration of $2.5*10^{11}$–$1*10^{13}$ GC/ml was injected, all viruses used are listed below (Table 1). Animals were imaged 2–4 weeks after injection and after a craniotomy was performed.

## Chronic and acute craniotomies

A head plate was glued to the skull and a craniotomy was made above the right somatosensory cortex. After dura removal the window was sealed with a glass coverslip. For injection of pharmacological blockers during imaging an acute craniotomy was performed and covered partially with a coverslip, while the surface was kept moist with aCSF (135 mM $Na^+$, 142.8 mM $Cl^-$, 4.2 mM $K^+$, 1 mM $Ca^{2+}$, 0.8 mM $Mg^{2+}$, 10 mM Glucose, 10 mM HEPES).

## Fluorescent dye-loading

MQAE (1(ethoxycarbonylmethyl)26-methoxyquinolinium bromide; Sigma-Aldrich, Munich, Germany[70], 7,5 mM in aCSF) was injected via bolus loading as described before[44] and via surface loading (30 min at RT). Astrocytes are co-labelled using SR-101 surface loading (100 μM, 1 min).

## Fibre photometry

A pair of excitation LEDs (465 nm and 405 nm, Doric Lenses, Tucker Davis Technologies) were connected to a minicube (Doric Lenses) by attenuator patch cords (400-μm core, NA = 0.48, Doric Lenses). The minicube contains dichroic mirrors and clean-up filters chosen to match the excitation and emission spectra. LEDs were controlled by LED drivers (Thorlabs, Doric, Tucker Davis Technologies) and connected to a RZ-5 or RZ10-X real-time processor (Tucker-Davis Technologies). 465 nm excitation light was delivered through a patch cord to stimulate mClY or EYFP fluorescence, respectively, while a recording with 405 nm excitation light was used to correct for bleaching and signal fluctuations due to movement. 465 nm/405 nm excitation were both sinusoidally modulated at 531 Hz/211 Hz. Fibre optic patch cords

(400 μm core, NA = 0.48, Doric Lenses) provided a light path between the minicubes and the animals. Zirconia sleeves were used to attach the fibre optic patch cords to fibre implants on the animal.

Each of the modulated signals generated by the LEDs were independently recovered using standard synchronous demodulation techniques implemented on the RZ-5/RZ10-X real-time processor (sampling rate of 1000 Hz). The commercial software Synapse (Tucker-Davis Technologies) was used to control the signal processor and was aligned to video and EEG/EMG signals through in- or outcoming TTL pulses. Files were exported for analysis to MATLAB (MathWorks) as described before[33]. ΔF/F calculations were based on the fitted 405 nm signal or by using the median of the fluorescence signal itself.

## In vivo macroscopic imaging

One-photon trans-craniotomy imaging was performed on awake head fixed mice, voluntarily running on an air-floating Styrofoam sphere, or immobilised in a MAG-1 mouse holder (Narishige, Japan). A self-build microscope (Cerna, Thorlabs) equipped with a 4x objective (RMS4X-PF, Olympus) was used. Fluorescence was recorded with a cooled EMCCD camera (Andor iXon Ultra897) at indicated frame rates. jRGECO and mClY were excited using a 580 nm and 470 nm LED (CoolLED pE-4000), respectively, and filtered by a dual band filter set (ET488/561x, ET488/561rdc, Chroma). Emitted light was first filtered with a 694 nm short pass filter to block the light from the behaviour LED (FF02-694/SP-25, Semrock) and further filtered with a 500 nm (ET500LP, Chroma) and 575 nm (AT575LP) long pass filter for mClY and jRGECO emission, respectively. Images were collected with μManager[71] (Version 2.0) and stored as 16-bit uncompressed tiff files.

## In vivo two-photon laser scanning microscopy for $Ca^{2+}$ imaging

2PM was performed on awake head-fixed mice, using a resonant scanner-based Bergamo B-scope (Thorlabs) with an InSight® DS + ™ laser (Spectra-Physics) with a 25x CFI plan apochromatic long working distance objective (Nikon, NA 1.10). GCaMP imaging was achieved at an excitation wavelength of 910 nm. Emission light was split by a 562 nm dichroic mirror (FF562-Di03, Semrock) and bandpass filtered by 525/50 nm (FF03, Semrock) and 607/70 nm (FF01, Semrock) into separate GaAsP PMTs. Images were acquired using ThorImage software at 30 Hz with a temporal averaging of 6 images to a final sampling rate of 5 Hz at a 512 ×512 pixel, 16-bit resolution.

## In vivo two-photon laser scanning microscopy for $Cl^-$ imaging

Intensity-based two-photon (2PM) chloride imaging was performed using a Galvo/Galvo scanner (Cambridge Technologies) equipped with

a water-immersion 20x objective (0.95 NA, Olympus). Fluorophores were excited using a Mai Tai DeepSee laser (SpectraPhysics). Images were acquired using Sciscan software at a frame rate of 3 Hz. Emitted fluorescence was recorded with GaAsP two-channel PMT (Scientifica Chromoflex). MQAE (exλ 770 nm/ emλ bp 460/50), mClY (exλ 960 nm/ emλ bp 525/50), SR101 (exλ 910 nm/ emλ bp 595/50).

## Fluorescence lifetime imaging microscopy (FLIM)
Emitted fluorescence was recorded with GaAsP single channel PMA hybrid detection unit (PicoQuant) connected by a light guide. TCSPC electronics (MultiHarp 150, PicoQuant) and acquisition software (SymPhoTime, PicoQuant) were used for fluorescence lifetime imaging as previously described[72]. Lifetime images were analysed using SymphoTime by fitting a bi-exponential function to the fluorescence decay. The average fluorescence lifetime of each astrocyte soma was calculated by manually drawing ROIs and averaging the values of all included pixels. All data represent mean ± SD. Calibration of MQAE chloride dependence was performed as described before[29,73].

## Brain state tracking and sleep scoring
Mice were placed in recording chambers (ViewPoint Behaviour Technology) and cables were connected to the EEG and EMG electrodes via a commutator (Plastics One, Bilaney). Mice were allowed to habituate to the recording chamber (ViewPoint Behaviour Technology) for at least one day/16–24 h prior to recordings. On the day of recording, mice were connected to the fibre optic implants and recordings were performed for 2–4 h. EEG and EMG signals were amplified (National Instruments Inc.) and filtered (EEG signal: high-pass at 1 Hz and low-pass at 100 Hz; EMG signal: high-pass at 10 Hz and low-pass at 100 Hz), and a notch filter of 50 Hz was used to reduce power line noise. Signals were digitised using a NI USB 6343 card (National Instrument) and sampled at a sampling rate of 512 Hz. Mouse behaviour was recorded continuously using an infra-red camera (Flir Systems) and used later to aid in the scoring of vigilance states. Hypnograms were created by visual inspection of EEG traces divided into 5 s and subsequently 1 s epochs. Vigilance states were defined as wake (high muscle tonus and a high frequency, low amplitude EEG), NREM sleep (no muscle tonus and low frequency, high amplitude EEG,), and REM sleep (no muscle tonus and high frequency, low amplitude EEG). Analysis of hypnograms was done using SleepScore software (4.0.0.40, ViewPoint Behaviour Technology). All data analysis was subsequently performed in MATLAB using custom-made scripts[33].

## Movement tracking of head fixed mice
Mouse behaviour was recorded at 25 Hz with a colour CCD camera (CS165CU1/M, Thorlabs) equipped with an 8 mm objective (MVL8M23, Thorlabs). During imaging the mouse was illuminated with a 780 nm LED (M780L3, Thorlabs). The infra-red filter of the camera was removed. Behaviour was analysed in DeepLabCut[74]. A model dataset of 1464 manually labelled frames from 14 individual recording sessions was used to train a ResNET-50 based network to recognise the centre of the front paws.

## Whisker stimulation
Neurones in the barrel-field cortex of the right hemisphere were stimulated 10 times by a series of air puffs (5 Hz, 50 ms, 20 psi) to contralateral whisker of the mouse over a time of 30 s with a break of 60 s between each trial. Only trials in which the mouse was not whisking or running during the 10 s before stimulation started were used for analysis.

## Optogenetic manipulation of astrocytic $[Cl^-]_i$
Optogenetic stimulation was performed using an external light source (Lumencor spectra X light, spectral output 100% = 310 mW/nm)

directed towards the craniotomy. NpHR is activated by yellow light (575 nm, continuous stimulation, 5 mW). SwiChR++ is opened by cyan (470 nm, 5 s, 5 mW) light and closed by red (600 nm, 5 s, 5 mW) light.

## Preparation of acute brain slices
Mice were anesthetised in a closed chamber with isoflurane (1.5%) and decapitated. Their brains were rapidly removed and immersed in an ice-cold cutting solution containing 230 mM sucrose, 2.5 mM KCl, 0.5 mM $CaCl_2$, 7 mM $MgSO_4$, 26 mM $NaHCO_3$, 1.2 mM $NaH_2PO_4$, 2 mM Myo-inositol, 2 mM sodium pyruvate, 0.4 mM ascorbic acid, 10 mM glucose, and saturated with 95% $O_2$ and 5% $CO_2$. Coronal slices (300 μm) were cut with a Leica VT1000S vibratome (Leica Biosystems, Buffalo Grove, IL, USA) and transferred to oxygenated artificial cerebrospinal fluid (aCSF) that contained 126 mM NaCl, 2.5 mM KCl, 2 mM $CaCl_2$, 2 mM $MgSO_4$, 26 mM $NaHCO_3$, 1.25 mM $NaH_2PO_4$, 10 mM glucose, and 10 mM lactate. Slices were incubated in aCSF for 1 to 5 h at room temperature before recording. During the recordings, the slices were placed in a perfusion chamber and superfused with aCSF gassed with 5% $CO_2$ and 95% $O_2$ at room temperature.

## Electrophysiology
Cells were visualised with a 40× water-immersion lens on an Olympus BX51 upright microscope (Olympus Optical Co., NY) equipped with differential inference contrast (DIC) optics. The cell images were viewed on a monitor with an infra-red differential interference contrast filter and a charge coupled device (CCD) camera (KP-M2RN, Hitachi, Japan). Areas with sufficient virus expression were identified through the eyepiece of the microscope using a Lumencor sola light engine for fluorescent illumination (NpHR3.0: EGFP exλ/emλ = 488/507 nm; SwiChR++: mTagBFP2 exλ/emλ = 402/457). Coronal sections with low or no expression were rejected.

Patch pipettes were fabricated using TW150F-4 glass capillaries (i.d. 1.12 mm, o.d. 1.5 mm, World Precision Instruments, USA) using a PC-10 electrode puller (Narishige International USA, Inc. East Meadow, NY, USA). The electrode resistance ranged from 4 to 5 MΩ. The patch pipette solution consisted of 130 mM CsCl, 1 mM $CaCl_2$, 2 mM $MgCl_2$, 10 mM Hepes-NaOH, 0.2 mM EGTA-KOH, 2.5 mM Na2ATP, 0.5 mM Na2GTP, and mM 5 QX-314. In these recordings, 6-cyano-7-nitroquinoxaline-2,3-dione (CNQX, 20 μM), D-(−)-2-amino-5-phosphonopentanoic acid (D-AP5, 50 μM) and CGP55845 (3 μM) were applied in the aCSF to block synaptic currents other than $GABA_A$ receptor-mediated currents.

Whole-cell patch clamp recordings of spontaneous IPSCs from cortical neurones were recorded in voltage clamp configuration with an Axopatch MultiClamp 700B amplifier (Axon Instruments), clamping the membrane potential at −60 mV. The recordings consisted of a 1 min baseline, 1 min photoactivation of NpHR3.0 and SwiChR++ using an external light source (CoolLED pE-4000) directed towards the recording chamber, and 1 min after activated channels were turned off. NpHR3.0 is activated by constant stimulation using light with a wavelength of 575 nm (50% of 150 mWatt). SwiChR++ is activated in response to 5 s light stimulation at a wavelength of 480 nm (50% of 150 mWatt) and deactivated by 5 s light stimulation at a wavelength of 635 nm (50% of 100 mWatt). In the presence of CNQX (20 μM), D-AP5 (50 μM) and CGP55845 (3 μM), tetanus stimulation (100 times at 100 pA, 200 μs, 50 Hz) was delivered at 10 s after starting the recording (Baseline) and again at 10 s before the end of channel activation (ON). Electrical stimulation was delivered at each holding potential, varying from −30 mV (SwiChR++) to −40 mV (NpHR3.0) through a constant isolated current source (an ISO-Flex isolator with a Master-8 stimulator; AMPI) with a concentric bipolar electrode placed in a distance of 100–150 μm from the recording cell. Current amplitudes were compared based on the ratio of early amplitudes (5–15 pulse) to late amplitudes (85–95 pulse). To study $GABA_A$ receptor antagonist-insensitive currents, recordings were made in the presence of

**Table 2 | The list of antibodies used in this study**

| Primary antibody | Company | Cat. No. |
|---|---|---|
| Anti-RFP antibody, 1:500 | Abcam | AB62341 |
| Mouse anti-neuronal nuclei (NeuN) monoclonal antibody, clone A60, 1:500 | Merck Millipore | MAB377 |
| Anti-Gfap antibody, 1:500 | Thermo Fisher Scientific | PA1-10004 |
| Anti-GFP antibody, 1:500 | Thermo Fisher Scientific | A-6455 |
| Anti-mCherry antibody, 1:500 | Thermo Fisher Scientific | M11217 |
| Secondary antibody | Company | Cat. No. |
| Goat-anti-chicken IgY (H + L) | Thermo Fisher Scientific | A21449 |
| Goat-anti-rabbit IgG (H + L) | Life Technologies/invitrogen | A11034 |
| Goat-anti-rabbit IgG (H + L) | Thermo Fisher Scientific | A11011 |
| Goat-anti-mouse-IgG2b | Thermo Fisher Scientific | A32728 |
| Goat-anti-rat IgG (H + L) | Abcam | AB1755710 |

200 μM picrotoxin (PTX) or at 0 mV holding potential. Recordings with drifting baselines over −200 pA were rejected. Series resistance was monitored during experiments, and recordings with changes over 20% of control during experiments were rejected.

All recordings were digitalised via a Digidata 1440A (Axon Instruments), and signals were filtered through a low-pass filter with a 2-kHz cut-off frequency and sampled by pCLAMP 10.2 software (Molecular Devices) with an interval of 50 μs. All analysis were performed using costume made MATLAB scripts.

### Chemicals
D-AP5 (79055-68-8), CNQX (115066-14-3), QX 314 chloride (5369-03-9), CGP55845 hydrochloride (149184-22-5), and Triton X-100 (9002-93-1) were from Tocris. MQAE (162558-52-3) Sigma Aldrich. All other chemicals used to prepare aCSF and pipette and slice cutting solution were from Sigma Aldrich.

### Data analysis
Fluorescence recordings were corrected for motion errors with the motion correction plugin of EZcalcium[75] and post-processed in Fiji[76]. A squared ROI was selected over the area with the highest intensity and the time-trace was saved. The mean fluorescence intensity uopn the whisker puff stimulation trials for each animal was calculated.

Motion-onset was calculated from the x and y position of the animals left front paw after DeepLabCut analysis in Matlab. The moment of movement-onset was defined as an event were the front paw displacement from one consecutive frame to the other was more than 150 pixels (=6.4 mm) after a phase of at least 10 s of still standing on the sphere. MQAE and mClY are quenched by Cl⁻. The fluorescence intensity is inversely correlated with surrounding [Cl⁻]. Therefore, all traces recorded from mClY, MQAE, YFP and autofluorescence were inverted.

### Drugs and pharmacology
Drugs were diluted in aCSF. Drug were used in the following concentration: GABA (56-12-2) 500 μM, Muscimol (2763-96-4) 500 μM, Diazepam (439-14-5) 1 mg/kg i.p.

### Immunohistochemistry
Brains were dissected and post-fixed in 4% PFA overnight and transferred to PBS until sectioning. Next, 100-μm sections were cut using a vibratome. Sections were then blocked in PBS with 5% goat serum and 0.3% Triton X-100 at room temperature for 1–2 h before overnight incubation with primary antibodies at 4 °C, all antibodies used are listed below (Table 2). After washing, sections were incubated with secondary antibodies at room temperature for 2 h. Images of brain slices were acquired using a Nikon Instruments Ni-E motorised microscope equipped with a ×4 CFI Plan Apo Lambda objective (0.2 NA). Images were collected with NIS-Elements (Nikon). For excitation, a halogen light source was used in combination with excitation filters 362–389 nm, 465–495 nm, 530–575 nm and Cy5 628–640 nm.

### Statistics and reproducibility
No statistical methods were used to pre-determine sample sizes, but our sample sizes are similar to those reported in previous publications[77-79]. Most data collection and analysis were not performed blinded to the conditions owing to the automatic nature of the experiments and analysis. The Shapiro–Wilk test was used to assess normality of data. Two-sided paired *t*-test or unpaired *t*-test was employed to compare pairs of groups, if data passed the normality test. Otherwise, the Wilcoxon matched-pairs signed-rank test or Mann–Whitney test was used for comparison. One-sample *t*-test was used to test whether population mean was different from a specific value. $*p < 0.05$, $**p < 0.005$, $***p < 0.001$, $****p < 0.0001$. Immunohistochemical experiments were repeated independently with similar results in up to three animals.

### Reporting summary
Further information on research design is available in the Nature Portfolio Reporting Summary linked to this article.

## Data availability
The data generated in this study are provided in the Source Data file. Raw data will be provided upon request. Source data are provided with this paper.

## Code availability
The custom-made MATLAB script used für IPSC analysis can be downloaded from Github: https://github.com/VUntiet/IPSC-analysis.

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

## Acknowledgements

We would like to thank the foundations for the generous funding they provided for this project: Lundbeck Foundation grant R287-2018-2046, V.U. Lundbeck Foundation grant R386-2021-165, M.N. Novo Nordisk Foundation grant NNF20OC0066419, M.N. National Institutes of Health grant R01AT011439, M.N. National Institutes of Health grant U19NS128613, M.N. US Army Research Office grant MURI W911NF1910280, M.N. Human Frontier Science Program grant RGP0036, M.N. The Dr. Miriam and Sheldon G. Adelson Medical Research Foundation, M.N. Simons Foundation grant 811237, M.N. Viral Vector Core, Gunma University Initiative for Advanced Research (GIAR) JP21dm0207111, H.H. We would like to thank Dan Xue for help with graphical design.

## Author contributions

V.U., A.V. and M.N. conceived the study and designed experiments. V.U., F.R.M.B., P.K., N.K., A.L., W.S., M.A., N.H., Z.B., B.S., S.D. and N.C.P. performed the experiments and collected the data. V.U., F.R.M.B., P.K., A.L., W.S., C.K., M.A., N.H., B.S., S.D., H.H. and N.C.P. performed the analysis. V.U. and A.V. wrote the manuscript, V.U., A.V. and M.N. edited the manuscript. All authors read and approved the final version of the manuscript.

## Competing interests

Authors declare no competing interests.
