## [Peer Review File · Nature Communications]

Astrocytic chloride is brain state dependent and modulates
inhibitory neurotransmissionREVIEWER COMMENTS

Reviewer #1 (Remarks to the Author):

This paper is very interesting and shows state-of-the-art results. Verena Untiet revealed that astrocytes store Cl⁻ and that Cl⁻ efflux from astrocytes contributes to modulate neural inhibition for the first time. I suggest additional experiments, if possible, and improvement of figures to further improve the quality of this excellent paper.

Major:

- 1) The authors demonstrated that Cl⁻ release from astrocyte through GABAA receptors (Figure 4). They, furthermore, showed that optogenetic increase in astrocyte [Cl⁻]_i attenuates locomotion-/whisker stimulation-induced neural [Ca²⁺]_i (Figure 3). However, it is still unclear whether Cl⁻ released from astrocytes flow into neurons via GABAA receptors, and then directly inhibits neural activity. To clarify this, I recommend that the authors record IPSCs using a slice patch clamp. Showing an increase in the amplitude of IPSCs during optogenetic increase in astrocyte [Cl⁻]_i would strongly support the authors' hypothesis.
- 2) The authors should validate that the expression is specific to astrocytes or neurons, the expression pattern of all experiments by showing immunohistochemistry staining. The position of tips of fiber optics in fiber photometry experiments should also show in figures. Histological data of negative control in which astrocytes express YFP is also required.
- 3) In Supplementary Figure 3F, an increase in astrocyte [Cl⁻]_i is shown after light illumination. There is, however, a lack of information about time. How long does the Cl⁻ concentration in astrocytes increase? Otherwise, it is difficult to discuss because of the lack of comparison with changes in neural Ca²⁺ concentrations.

Minor:

- 1) It is very interesting that astrocyte [Cl⁻]_i altered according to sleep/wakefulness state. Cl⁻ level is high during sleep. Previous studies have demonstrated that neural activity in the cortex is inhibited during sleep compared to wakefulness. Please discuss this relationship.
- 2) I am wondering why the Cl⁻ concentration transiently increases just before it drops. Is there any possibility involving the Cl⁻ pump since the electrochemical gradient does not explain it? Please discuss it.
- 3) Figure 1E. Astrocyte [Cl⁻]_i increases immediately before the transition from wakefulness to sleep. Is it possible that a change in concentration induces sleep?
- 4) Line 70. Please cite the original paper of mClY.
- 5) Legend of Figure 1. (I) is not mentioned.
- 6) Line 96. I could not catch how to define mobile and immobile here. Please explain it in the main text.
- 7) Supplementary Figure 2. Are the label of vertical axes correct? Isn't it "Cl⁻ vs power" instead of "sleep/wake vs power"? Please check it?
- 8) Supplementary Figure 2. Have you performed statistical processing?
- 9) Figure 2D and 2G. These two panels should have the same X-axis for easy comparison.
- 10) Supplementary Figure 3. I could not follow how MQAE express specifically in astrocyte.
- 11) Line 199. "mic" should be "mice."
- 12) Line 200. "Cl⁻I" should be "[Cl⁻]_i."
- 13) Line 235. "Figure 4C" should be "Figure 4C and D."
- 14) Line 235. "Figure 4D" should be "Figure 4E."

Reviewer #2 (Remarks to the Author):

This is an interesting and innovative study that focuses on a novel and potentially

important issue, the role of astroglial chloride in regulating inhibitory activity of brain networks. The authors employ a battery of state-of-the-art neuroscience techniques to test their hypothesis that the GABA_A receptor-dependent efflux of chloride from cortical astrocytes could boost inhibitory transmission in the region. There are several important technical issues that the authors are encouraged to address.

1. The mCIY fluorescent recordings presented in Figure 1 display significant fluctuations yet it would be important to understand how this reflects the changes in astrocyte Cl⁻ concentration, at least in relative terms. Supplementary Fig. 3 data claim but do not seem to present Cl⁻ concentrations. A more definite concentration calibration test would be helpful.

2. Figure 3 data as presented require further explanations and additional controls, which would be important to the concept overall.

(i) Optogenetic light stimulation is known to produce significant side effects, mainly due to local brain tissue heating (Studjenski et al Cell Rep 2015; Owen et al Nat Neurosci 2019) and associated physiological concomitants. This requires some control.

(ii) Bulk changes in neuronal activity, as reported by the Ca²⁺ sensor (under hSyn1 promoter) here, might reflect decreased activity of either interneurons or principal neurons, or relative changes in both groups, which is difficult to interpret.

(iii) It is not clear if astrocyte Cl⁻ traces shown in C and G panels are recorded in control conditions or under optogenetic stimulation of NpHR3.0. In any case, one expects illustrations for both conditions.

3. Supplementary Fig. 3G data indicate that Halorhodopsin activation can increase astrocyte chloride by ~10% (2-4 mM), on average. While this change might well trigger some physiological response from astrocytes, the authors should explain how it could have a significant effect on inhibitory synapses directly, by altering extracellular ion homeostasis as hypothesized, assuming ~120 mM extracellular Cl⁻.

4. GABA_AR receptors have been found in Bergmann glia, in the cerebellum. The authors should present and discuss evidence for a significant presence of GABA_AR in cortical astrocytes.

5. Fig. 4 data require further clarification and controls.

(i) Muscimol application will activate GABA_ARs at multiple cell targets, including principal neurons and interneurons, which in turn could affect astrocyte activity. This makes it difficult to interpret the data as a direct effect of astrocyte activity.

(ii) Bolus loading with MQAE would label all cells in the region including neuronal elements intermingled with SR101 labelled astroglia. Distinguishing between astroglial and local neuronal MQAE signals would not appear feasible in these conditions.

Reviewer #3 (Remarks to the Author):

The manuscript by Unitiet, et al., reports that the astrocytic Cl⁻ is brain state dependent and it regulates cortical neuronal activity. Authors employed very challenging in vivo fiber photometry from awake, behaving mouse brain, imaging intracellular astrocytic Cl⁻. In addition, authors employed NpHR to artificially manipulate the intracellular Cl⁻ upon light stimulation to see if the neuronal activity is altered. Although the authors should be praised for heroic attempt to record astrocytic Cl⁻ and manipulate astrocytic Cl⁻, there are several major concerns that need to be addressed to provide convincing evidence for their claims.

1. In figure 1 authors found astrocytic Cl⁻ is dramatically changed during sensory input or motor activity in behaving mouse brain. Authors used a genetically encoded Cl⁻ sensor mCIY, which is a YFP variant that is supposed to be pH-independent. Other predecessor Cl⁻ sensors such as YFP, Clomeleon, and SuperClomeleon all suffer from pH-sensitivity. Authors in this study used YFP as a negative control. Authors found that there was no change with YFP. This negative result with GFAP-YFP is odd because if the positive mCIY signal was due to a bona fide Cl⁻, YFP should also give a positive signal for YFP is also a Cl⁻ sensor. This negative data by YFP can be interpreted as the positive signal by mCIY is not due to Cl⁻ but rather something other than Cl⁻ or pH.
2. In figure 2D, G, H, authors report a biphasic change in mCIY during movement and whisker stimulation. This biphasic change is reminiscent of the previous report by Woo, et al., 2020, J of Physiology, in which the similar biphasic change of Cl⁻ and its detailed molecular mechanism was described in hippocampus CA1. In particular, the initial increase in mCIY, a putative influx of Cl⁻, is very interesting and this was previously demonstrated to be mediated possibly by GlyT1 (Woo, et al., 2020). However, authors did not pay attention to this initial increase but rather focused on the decreasing phase of mCIY. The decrease of mCIY, a putative efflux of Cl⁻, is also reminiscent of the Cl⁻ efflux reported by Woo, et al, 2020, in which the Cl⁻ efflux was demonstrated to be mediated by Best1 (Woo, et al., 2020). Authors should cite this paper and discuss about the possible molecular mechanism of this biphasic response of mCIY.
3. While the authors were focusing on the Cl⁻ efflux phase, they try to mimic the influx of Cl⁻ by employing NpHR, which will increase the intracellular Cl⁻. Thus, this employment of NpHR does not fit very well with their focus on the efflux of Cl⁻ from astrocytes to provide extracellular Cl⁻ for supply to neurons. What the authors really need is a manipulation that mimics efflux of Cl⁻. Therefore, it is unfortunate that there is a disconnect between what they really want to mimic and what they ended up using.
4. In figure 4, authors demonstrate that GABA can induce a decrease in Cl⁻ in astrocytes. This is a useful addition. However, this needs to be further expanded to address the real question of whether the putative Cl⁻ efflux in behaving animal during sensory stimulation or motor activity is due to the astrocytic GABA_A receptors. Cell-type specific manipulation (gene silencing or cKO) of GABA_A receptor subunit in astrocyte is needed.
5. Authors claim that astrocytes serve as a source of Cl⁻ required for sustained GABAergic transmission. However, authors do not provide direct evidence for this. Instead, they provide indirect lines of evidence that the increase in Cl⁻ influx by NpHR activation in astrocytes causes a reduced neuronal activity. There is a huge gap between increase in Cl⁻ influx in astrocytes and providing Cl⁻ required for sustained GABAergic transmission in neurons. Authors need to fill in the missing gap with additional evidence. Otherwise the claim needs to be toned down.
6. Authors claim that NpHR-induced Cl⁻ influx leads to Cl⁻ efflux that increases [Cl⁻]_o. However, authors do not provide evidence for this. The [Cl⁻]_o is already very high, at around 130 mM. It remains unclear how much astrocytic Cl⁻ efflux will increase this already high [Cl⁻]_o, unless authors directly measure the [Cl⁻]_o to demonstrate the increases in [Cl⁻]_o.
7. It is unclear that the collapse of the neuronal Cl⁻ gradient during intense GABA_A receptor activity is due to an increase in Cl⁻ accumulation or decrease in [Cl⁻]_o. Because of very high concentration of 130 mM [Cl⁻]_o, it is more likely that the collapse is due to an increase in Cl⁻ accumulation. Authors need to clarify this assumption.

Minor comments

1. Line 194: accelerates is misspelled
2. Line 199: mice is misspelled
3. Line 245: "lasted for 15.10 ± 10.1" needs a unit.
4. Line 255: that -> than
5. Line 283: "-0.)"?

REVIEWER COMMENTS

Reviewer #1 (Remarks to the Author):

This paper is very interesting and shows state-of-the-art results. Verena Untiet revealed that astrocytes store Cl^- and that Cl^- efflux from astrocytes contributes to modulate neural inhibition for the first time. I suggest additional experiments, if possible, and improvement of figures to further improve the quality of this excellent paper.

Major:

1) The authors demonstrated that Cl^- release from astrocyte through GABAA receptors (Figure 4). They, furthermore, showed that optogenetic increase in astrocyte $[\text{Cl}^-]_i$ attenuates locomotion-/whisker stimulation-induced neural $[\text{Ca}^{2+}]_i$ (Figure 3). However, it is still unclear whether Cl^- released from astrocytes flow into neurons via GABAA receptors, and then directly inhibits neural activity. To clarify this, I recommend that the authors record IPSCs using a slice patch clamp. Showing an increase in the amplitude of IPSCs during optogenetic increase in astrocyte $[\text{Cl}^-]_i$ would strongly support the authors' hypothesis.

As suggested, we recorded IPSC in acute cortical slices upon electrical stimulation as described before¹⁻³. In controls, the IPSC of the 90th stimulus (average of 85-95th stimulus) is smaller compared to the 10th stimulus (average of 5-15th stimulus) which reflects the collapse of the Cl^- gradient⁴⁻⁶. Activation of NpHR in astrocytes increased baseline astrocytic $[\text{Cl}^-]_i$ results in a slower run-down of the IPSCs. The higher 90th stimulus indicates a longer maintenance of the Cl^- gradient. In contrast, the optogenetic tool SwiChR, which is switchable Cl^- channel decreases baseline $[\text{Cl}^-]_i$ in astrocytes. Activation of SwiChR increases the ratio between 10th and 90th response indicative for a faster rundown of the response to stimulation. Thus, in response to the reviewer we now show that manipulation of astrocytic $[\text{Cl}^-]_i$ in both direction controls the strength of IPSC during prolonged excitation. These observations support the conclusion that astrocytic $[\text{Cl}^-]_i$ serves a source of Cl^- during prolonged neuronal activity by maintenance the Cl^- gradient. Supplemental Figure 7 was added to the manuscript.

Supplemental Figure 7: Optogenetic manipulation of baseline astrocytic $[\text{Cl}^-]_i$ affects activity-dependent depolarizing shifts of IPSC.

(a) IPSC was recorded in acute cortical slices upon electrical stimulation as described before¹. (b) In controls, the IPSC of the 90th stimulus (average of 85-95th stimulus) is smaller compared

to the 10th stimulus (average of 5-15th stimulus), which reflects the collapse of the Cl gradient. In contrast, activation of NpHR in astrocytes causing an increase in baseline astrocytic [Cl⁻]_i results in a slower run-down of the IPSCs. The higher 90th stimulus indicates a longer maintenance of the Cl⁻ gradient. (c) To study GABA_A receptor antagonist-insensitive currents, recordings were made in the presence of 200 μM picrotoxin (PTX) or at 0 mV holding potential. (d) The optogenetic tool SwiChR, which is switchable Cl⁻ channel decreases baseline [Cl⁻]_i in astrocytes. Activation of SwiChR increases the ratio between 10th and 90th response indicative of a faster rundown of the response to stimulation. (N = 4-5 mice). All recordings were performed in presence of CNQX (20 μM), D-AP5 (50 μM) and CGP55845 (3 μM), tetanus stimulation (100 times at 100 pA, 200 μs, 50 Hz). Data represent mean ± SEM.

2) The authors should validate that the expression is specific to astrocytes or neurons, the expression pattern of all experiments by showing immunohistochemistry staining. The position of tips of fiber optics in fiber photometry experiments should also show in figures. Histological data of negative control in which astrocytes express YFP is also required.

We thank the reviewer for pointing this out. We have revised the manuscript by adding Supplemental Figure 8, showing the cell-specific expression of sensors and the position of the fiber optics for fiber photometry recordings.

Supplemental Figure 8: Validation of cell specificity of biosensor expression.

(a) Position of the tip of the fiber optics used for fiber photometry. (b) Expression of Gfap-mCIY and Gfap-YFP in cortical astrocytes (green). (c) Expression of Gfap-eGFP, Gfap-mCIY or Gfap-NpHR3.0 (green) together with hsyn-jRGECO (red). Nuclei are labeled with DAPI (blue). Expression of Thy1-Gcamp (green) and Gfap-SwiChR (blue) (d and e) Colocalization of cell specific marker (astrocytes = Gfap or neurons = NeuN) and injected biosensor or optogenetic tool.

3) In Supplementary Figure 3F, an increase in astrocyte [Cl⁻]_i is shown after light illumination. There is, however, a lack of information about time. How long does the Cl⁻ concentration in astrocytes increase? Otherwise, it is difficult to discuss because of the lack of comparison with changes in neural Ca²⁺ concentrations.

Thank you for pointing this out. We agree that the comparison of the timing with neuronal Ca²⁺ is difficult, based on Supplemental Figure 3f. Using NpHR3.0-mCherry and mCIY on the microscope we were able to record the response to whisker stimulation. It shows that the prolonged decrease of astrocytic [Cl⁻]_i is reduced, having the biggest impact on the early phase (first 6 – 8 seconds) of Cl⁻ decrease. It has the same time course as the reduced neuronal Ca²⁺ response. New recordings are illustrated in Figure 3.

Figure 3: Optogenetic elevation of astrocytic $[Cl^-]_i$ shortens activation-induced neuronal $[Ca^{2+}]_i$. (a) Experimental protocol: The optogenetic Cl^- pump was expressed in astrocytes to manipulate $[Cl^-]_i$, while neuronal $[Ca^{2+}]_i$ was imaged using jRGECO1a. As the negative control light stimulation was applied to mice not expressing NpHR3.0. The awake, head-fixed mice were voluntarily running on a Styrofoam sphere. (b) Average astrocytic $[Cl^-]_i$ trace during transition from stationary to mobile, while simultaneously stimulating the optogenetic tool NpHR3.0 in astrocytes, shading indicates SEM. (N = 5-6 mice). (c) Shows maximal (peak) amplitude of astrocytic $[Cl^-]_i$ upon movement onset; time to peak of astrocytic $[Cl^-]_i$ upon movement onset; recovery time to baseline; slope of decay of $[Cl^-]_i$ transient, and area under curve from 6-8 seconds (N = 5-6 mice). (d) Average neuronal $[Ca^{2+}]_i$ trace during transition from stationary to mobile, while simultaneously stimulating the optogenetic tool NpHR3.0 in astrocytes, shading indicates SEM. (N = 3 mice). (e) Shows maximal (peak) amplitude of neuronal $[Ca^{2+}]_i$ upon movement onset; time to peak of neuronal Ca^{2+} upon movement onset; recovery time to baseline; slope of decay of $[Ca^{2+}]_i$ transient (N = 5-6 mice). (f) Using the same protocol as in Figure 2, whiskers were stimulated using air puffs. (g) astrocytic $[Cl^-]_i$ trace during whisker stimulation in control and upon light-activation of optogenetic tool NpHR3.0 in astrocytes, shading indicates SEM. (N = 4-5 mice). (h) Maximal (peak) amplitude of neuronal $[Ca^{2+}]_i$ upon whisker stimulation; time to peak of astrocytic $[Cl^-]_i$ upon whisker stimulation. Average astrocytic $[Cl^-]_i$ during the period of maximal changes upon whisker stimulation, 6 – 8 sec after onset of stimulation. Area under the curve (AUC) of astrocytic $[Cl^-]_i$ during 6 – 8 sec after onset of stimulation. (i) Average neuronal $[Ca^{2+}]_i$ trace during whisker

stimulation in control and upon light-activation of optogenetic tool NpHR3.0 in astrocytes, shading indicates SEM. (N = 8-10 mice). (k) Maximal (peak) amplitude of neuronal $[Ca^{2+}]_i$ upon whisker stimulation; time to peak of neuronal $[Ca^{2+}]_i$ upon whisker stimulation. Neuronal $[Ca^{2+}]_i$ during the period of maximal astrocytic $[Cl^-]_i$ changes upon whisker stimulation, 6 – 8 sec after onset of stimulation. Area under the curve (AUC) of neuronal $[Ca^{2+}]_i$ during 5 sec after peak. Un-paired two-tailed t-test. Data represent mean \pm SEM.

Minor:

1) It is very interesting that astrocyte $[Cl^-]_i$ altered according to sleep/wakefulness state. Cl^- level is high during sleep. Previous studies have demonstrated that neural activity in the cortex is inhibited during sleep compared to wakefulness. Please discuss this relationship.

We agree and have updated the discussion by adding the following paragraph, page 24-25, line 482-507:

“Astrocytic Cl^- is regulated by brain state

Previous studies estimated astrocytic $[Cl^-]_i$ to range between 29 and 51 mM, depending on brain region, age and experimental setup^{27,33,34 28,29 30-32}. Yet, hitherto astrocytic $[Cl^-]_i$ in awake behaving mice was not measured. We here report that in vivo astrocytic $[Cl^-]_i$ is highly dynamic and depends on the brain state. During sleep or rest, when astrocytic $[Cl^-]_i$ is high and stable, astrocytic Ca^{2+} activity is low and cortical network patterns appear synchronised. A stereotypical pattern of dynamic changes in astrocytic $[Cl^-]_i$ is observed upon arousal and locomotion when cortical networks are desynchronised. A fast increase in astrocytic $[Cl^-]_i$ is followed by a prolonged decrease which terminates with the respective stimulus. Also, astrocytic $[Cl^-]_i$ is increasing before transitioning to sleep and remains high during NREM sleep. Previous studies described neural activity in cortex to be lower during sleep compared to wakefulness⁵⁹. Thus, the demand for astrocytic Cl^- release is low, possibly contributing to an increase in astrocytic $[Cl^-]_i$ during sleep. Future studies must address whether the changes in astrocytic Cl^- or causally involved in state dependent changes in neural activity or merely a result of alterations neural activity patterns. A recent publication shows that the equilibrium potential of GABA (E_{GABA}) in pyramidal neurones is dependent on the time of day, being hyperpolarised during sleep and becoming more depolarised during wakefulness. Reversal potential of GABA_A receptors (E_{GABA_A}) depends on neuronal transmembrane chloride gradient defined by $[Cl^-]_i$ and $[Cl^-]_o$ ⁶⁰. The study also shows that following sleep deprivation expression of NKCC1, and hence Cl^- accumulation into neurones, is upregulated and that this effect can be reversed by blocking NKCC1 using bumetanide⁶⁰. At the same time E_{GABA_A} also depends on $[Cl^-]_o$ in the synaptic cleft regulated by astrocytic $[Cl^-]_i$. High astrocytic $[Cl^-]_i$ during sleep provides Cl^- to maintain a E_{GABA_A} at hyperpolarised levels, whereas lower $[Cl^-]_i$ during wakefulness restricts Cl^- availability and results in depolarising shift of E_{GABA_A} . Hence, astrocytic $[Cl^-]_i$ participates in modulating neuronal E_{GABA_A} in a brain state dependent manner and must be taken into consideration.”

2) I am wondering why the Cl^- concentration transiently increases just before it drops. Is there any possibility involving the Cl^- pump since the electrochemical gradient does not explain it? Please discuss it.

There might be two possible explanations for this. As the reviewer suggested, it could be triggered by the rise of extracellular K^+ that is taken up by astrocytes together with Cl^- via

NKCC1. It could also be attributed to the activation of the glycine transporter GlyT1 as suggested before⁷. We addressed the reviewers comment by adding the following paragraph to the discussion, page 25, line 508-525:

“Astrocytic [Cl⁻]_i changes upon neuronal activity

Instigating neuronal activity by whisker stimulation or movement onset consistently triggered a biphasic response of astrocytic [Cl⁻]_i (Figure 2). Fast increase in astrocytic [Cl⁻]_i could be explained by Cl⁻ influx through, for example, the glycine transporter GlyT1, which is reported to have the same time course as K⁺ uptake through K2P channels⁷. The transporter NKCC1 can be potentially involved too, yet NKCC1 is not expressed in astrocytes in the adult forebrain^{8,9}. Volume changes mediated by increases in extracellular K⁺ are unlikely to trigger a transient increase in [Cl⁻]_i, as astrocytic swelling upon physiological increases of K⁺ develops within much slower time range of minutes¹⁰. Prolonged decrease of astrocytic [Cl⁻]_i is likely mediated by opening of GABA_AR, since puff application of GABA or muscimol triggers a similar Cl⁻ release from astrocytes (Figure 5). Electrical stimulation of Schaffer collateral pathway results in the prolonged decrease in astrocytic [Cl⁻]_i mediated by Best-1 Cl⁻ channel in brain slices⁷, but the observed slow decay (70-80 seconds) is several fold longer than the 10-15 seconds observed in our experiments (Figure 2). Evoked astrocytic Ca²⁺ transients start a few seconds after stimulus onset and last much longer than the stimulation¹¹. In comparison to that the here described astrocytic Cl⁻ transients, respond immediately upon initiation of the stimulation, and terminate quickly after the stimulation stops. This indicates that astrocytic Cl⁻ and Ca²⁺ transients are not directly correlated.”

3) Figure 1E. Astrocyte [Cl⁻]_i increases immediately before the transition from wakefulness to sleep. Is it possible that a change in concentration induces sleep?

Thank for pointing this out. The brain state dependence of astrocytic [Cl⁻]_i is very interesting and knowing the GABAergic inhibition plays a crucial role in sleep we are looking forward to further investigating this topic. However, it has been shown that sleep and wakefulness are controlled by sleep or wake promoting nuclei including locus coeruleus (LC)¹². Activity of LC followed by norepinephrine release is linked to wakefulness while reduced activity of LC promotes sleep¹³. We address the comment in the discussion, page 24, lines 490-496.

“Also, astrocytic [Cl⁻]_i is increasing before transitioning to sleep and remains high during NREM sleep. Previous studies described neural activity in cortex to be lower during sleep compared to wakefulness⁵⁹. Thus, the demand for astrocytic Cl⁻ release is low, possible contributing to an increase in astrocytic [Cl⁻]_i, during sleep. Future studies must address whether the changes in astrocytic Cl⁻ or causally involved in state dependent changes in neural activity or merely a result of alterations neural activity patterns.”

4) Line 70. Please cite the original paper of mCIY.

This was an oversight. We have added the reference.

5) Legend of Figure 1. (I) is not mentioned.

Thanks for highlighting this, we have fixed the omission by adding the missing part of the figure legend:

“(I) Experimental Protocol: Cortical astrocytic [Cl⁻]_i was imaged in awake or resting (immobile) mice, as well as in voluntary running (mobile) mice.”

6) Line 96. I could not catch how to define mobile and immobile here. Please explain it in the main text.

We apologise for not being clearer about the definition of mobile and immobile states during wakefulness. The manuscript was revised, page 5-6, line 116-125:

“To analyse the impact of spontaneous locomotion on $[Cl^-]_i$, periods of mobile and immobile awake states were quantified by tracking videos of home cage mobility using EthoVision XT (Noldus). Overall, mobile and immobile periods were defined¹⁴ as follows: (i) intervals below 5% change in body area were detected as immobility, whereas (ii) intervals above 5% change in body area were considered as mobility periods. Transitions that were preceded by a minimum of 10 s immobility and with mobile periods lasting at least 1 s were chosen for assessment of mClY fluctuations (Figure 1k). A biphasic (short increase followed by a prolonged decrease) $[Cl^-]_i$ transient was consistently associated with the onset of locomotion (Figure 1l).”

7) Supplementary Figure 2. Are the label of vertical axes correct? Isn't it “Cl- vs power” instead of “sleep/wake vs power”? Please check it?

Thank you, the label was misleading. We meant Cl^- during awake and Cl^- during wakefulness. The label was corrected.

8) Supplementary Figure 2. Have you performed statistical processing?

Thanks for pointing this out, the statistical evaluation was added to the figure description of Supplemental Figure 2:

“The average Pearson correlation coefficient: (a) delta: $r=0.01$, theta $r= 0.1$, beta $r=0.16$, sigma $r=0.17$ (b) delta $r=0.18$, theta $r=0.45$, beta $r= 0.06$, sigma $r=0.01$.”

9) Figure 2D and 2G. These two panels should have the same X-axis for easy comparison.

We agree with the reviewer the x-axis of Figure 2d was extended to 60 second to match the x-axis of 2Fg.

10) Supplementary Figure 3. I could not follow how MQAE express specifically in astrocyte.

Thanks for pointing out that our description was misleading. MQAE is not genetically expressed but a fluorescent dye. It is an AM-Ester that penetrates the cellular membrane by diffusion. Inside the cell it gets cleaved by esterase and becomes membrane impermeable. The dye accumulates in all cell-types; however, it accumulates in a higher amount in astrocytes compared to neurons. MQAE was imaged on the 2-PM microscope and SR-101 was used to label astrocytes. Astrocyte somata were analyzed. The figure legend was updated by adding the information that MQAE is a fluorescent dye. The method section includes a detailed description about fluorescent dye loading.

11) Line 199. “mic” should be “mice.”

Thanks for finding this typo, it was corrected.

12) Line 200. “Cl-I” should be “[Cl-i].”

Thanks for pointing this out, it was corrected.

13) Line 235. “Figure 4C” should be “Figure 4C and D.”

We thank the reviewer for finding this oversight. The error was corrected.

14) Line 235. “Figure 4D” should be “Figure 4E.”

We thank the reviewer for pointing out this mix up. The error was corrected.

Reviewer #2 (Remarks to the Author):

This is an interesting and innovative study that focuses on a novel and potentially important issue, the role of astroglial chloride in regulating inhibitory activity of brain networks. The authors employ a battery of state-of-the-art neuroscience techniques to test their hypothesis that the GABAA receptor-dependent efflux of chloride from cortical astrocytes could boost inhibitory transmission in the region. There are several important technical issues that the authors are encouraged to address.

1. The mCIY fluorescent recordings presented in Figure 1 display significant fluctuations yet it would be important to understand how this reflects the changes in astrocyte Cl⁻ concentration, at least in relative terms. Supplementary Fig. 3 data claim but do not seem to present Cl⁻ concentrations. A more definite concentration calibration test would be helpful.

Thanks for making this point, we share this opinion with the reviewer. It would be ideal to know the absolute [Cl⁻]_i in mM *in vivo*, both baseline and after optogenetic manipulation. However, we cannot use calibration curves obtained in brain slices or cuvette experiments to translate to the *in vivo* data. In Supplemental Figure 3, the fluorescence lifetime was recorded before and after optogenetic stimulation, fluorescence lifetime is an absolute measurement of Cl⁻ and we can conclude that the [Cl⁻]_i increases. The calibration in brain slices which we performed cannot be used to draw a conclusion from the *in vivo* readings. However, it gives the hint that astrocytic [Cl⁻]_i is above diffusion equilibrium, the field tried to work around this limitation, also see¹⁵. We added Supplemental Figure 9 showing the calibration of MQAE in brain slices, including an interpretation of the values recorded *in vivo*. To get an impression with regard to the relative change of astrocytic Cl⁻ we added Cl⁻ imaging to Figure 3, which shows how astrocytic [Cl⁻]_i changes upon whisker stimulation or movement onset, when manipulated with optogenetics. This highlights that the biggest change of [Cl⁻]_i occurs at the same time as we observe change in neuronal signaling.

Supplemental Figure 9: Calibration of MQAE fluorescence lifetime in brain slices.

(a) Stern Vomer plot of the calibration of Cl⁻ dependence of MQAE fluorescence lifetimes in acute brain slices. (b) Translation of *in vivo* recorded fluorescence lifetimes into absolute Cl⁻ concentration based on the *in situ* calibration curve. (c) Color coded Fluorescence lifetime images of the different Cl⁻ concentration calibrated. (N = 4 mice). Data represent mean ± SEM. Limitation of the calibration: Cl⁻ sensitivity of MQAE is pH independent and not affected by other ions, however dye hydrolysis, self-quenching and non-specific quenching by cytosolic proteins are cell type and sample specific. Therefore, the calibration performed in acute brain slices cannot reliably translate recordings obtained *in vivo*¹⁶⁻¹⁸.

2. Figure 3 data as presented require further explanations and additional controls, which would be important to the concept overall.

We thank the reviewer for the following feedback and addressed the individual point of critique in detail below:

(i) Optogenetic light stimulation is known to produce significant side effects, mainly due to local brain tissue heating (Studjenski et al Cell Rep 2015; Owen et al Nat Neurosci 2019) and associated physiological concomitants. This requires some control.

We agree with the reviewer, light is a source of energy which can harm the tissue and cause side effects. To ensure the observed effect to be mediated by Cl^- and not being an artefact due to phototoxicity, all data which was collected with optogenetic stimulation are compared with a control. These controls were imaged and stimulated with the same light sources and settings (3-5 mWatt) as used for optogenetic stimulation. All traces in Figure 3 were recorded with active stimulation light, while the control did not express NpHR3.0. The experiment could be performed repeatedly with the same signal and without observing damage to the tissue. The new recordings obtained upon SwiChR expression were compared to control recording without light stimulation as well as control recordings with nonspecific light stimulation. Furthermore, we show in Supplemental Figure 4 that light stimulation without expression of optogenetic tools does not affect LFP recordings.

(ii) Bulk changes in neuronal activity, as reported by the Ca^{2+} sensor (under hSyn1 promoter) here, might reflect decreased activity of either interneurons or principal neurons, or relative changes in both groups, which is difficult to interpret.

Thank you for pointing this; to improve our experimental setting, we used Thy1-Gcamp mice which express the Ca^{2+} sensor only in excitatory neurons for the additional experiments. Concerning the data which had already been collected with a Ca^{2+} expressed under the *hsyn* promoter we argue that this is still valid. Of all neurons in cortex only 11.5% are interneurons¹⁹. Therefore, we conclude that 90% of signal we image represents activity of excitatory neurons. We agree an opposite effect on the activity of interneurons will change the overall signal, however the effect is minor and can be neglected.

(iii) It is not clear if astrocyte Cl^- traces shown in C and G panels are recorded in control conditions or under optogenetic stimulation of NpHR3.0. In any case, one expects illustrations for both conditions.

We thank the reviewer for pointing this out. We have revised Figure 3, by adding recordings of astrocytic $[\text{Cl}^-]_i$ with and without optogenetic stimulation of NpHR3.0.

Figure 3: Optogenetic elevation of astrocytic $[Cl^-]_i$ shortens activation-induced neuronal $[Ca^{2+}]_i$. (a) Experimental protocol: The optogenetic Cl^- pump was expressed in astrocytes to manipulate $[Cl^-]_i$, while neuronal $[Ca^{2+}]_i$ was imaged using jRGECO1a. As the negative control light stimulation was applied to mice not expressing NpHR3.0. The awake, head-fixed mice were voluntarily running on a Styrofoam sphere. (b) Average astrocytic $[Cl^-]_i$ trace during transition from stationary to mobile, while simultaneously stimulating the optogenetic tool NpHR3.0 in astrocytes, shading indicates SEM. (N = 5-6 mice). (c) Shows maximal (peak) amplitude of astrocytic $[Cl^-]_i$ upon movement onset; time to peak of astrocytic $[Cl^-]_i$ upon movement onset; recovery time to baseline; slope of decay of $[Cl^-]_i$ transient, and area under curve from 6-8 seconds (N = 5-6 mice). (d) Average neuronal $[Ca^{2+}]_i$ trace during transition from stationary to mobile, while simultaneously stimulating the optogenetic tool NpHR3.0 in astrocytes, shading indicates SEM. (N = 3 mice). (e) Shows maximal (peak) amplitude of neuronal $[Ca^{2+}]_i$ upon movement onset; time to peak of neuronal Ca^{2+} upon movement onset; recovery time to baseline; slope of decay of $[Ca^{2+}]_i$ transient (N = 5-6 mice). (f) Using the same protocol as in Figure 2, whiskers were stimulated using air puffs. (g) astrocytic $[Cl^-]_i$ trace during whisker stimulation in control and upon light-activation of optogenetic tool NpHR3.0 in astrocytes, shading indicates SEM. (N = 4-5 mice). (h) Maximal (peak) amplitude of neuronal $[Ca^{2+}]_i$ upon whisker stimulation; time to peak of astrocytic $[Cl^-]_i$ upon whisker stimulation. Average astrocytic $[Cl^-]_i$ during the period of maximal changes upon whisker stimulation, 6 – 8 sec after onset of stimulation. Area under the curve (AUC) of astrocytic $[Cl^-]_i$ during 6 – 8 sec after onset of stimulation. (i) Average neuronal $[Ca^{2+}]_i$ trace during whisker stimulation in control and upon light-activation of optogenetic tool NpHR3.0 in astrocytes,

shading indicates SEM. (N = 8-10 mice). (k) Maximal (peak) amplitude of neuronal $[Ca^{2+}]_i$ upon whisker stimulation; time to peak of neuronal $[Ca^{2+}]_i$ upon whisker stimulation. Neuronal $[Ca^{2+}]_i$ during the period of maximal astrocytic $[Cl^-]_i$ changes upon whisker stimulation, 6 – 8 sec after onset of stimulation. Area under the curve (AUC) of neuronal $[Ca^{2+}]_i$ during 5 sec after peak. Un-paired two-tailed t-test. Data represent mean \pm SEM.

3. Supplementary Fig. 3G data indicate that Halorhodopsin activation can increase astrocyte chloride by ~10% (2-4 mM), on average. While this change might well trigger some physiological response from astrocytes, the authors should explain how it could have a significant effect on inhibitory synapses directly, by altering extracellular ion homeostasis as hypothesized, assuming ~120 mM extracellular Cl^- .

We understand the reviewer's concern and would like to highlight our data as well as studies performed by others to argue for our hypothesis.

First, we cannot translate the changes in $[Cl^-]_i$ recorded in Supplemental Figure 3g in mM changes, thus we cannot conclude that 10% correlates to a 2-4 mM changes. Furthermore, we did record changes of astrocytic $[Cl^-]_i$ as recording $[Cl^-]_o$ in the synaptic cleft is currently not feasible. Therefore, we can only speculate about the absolute mM changes of $[Cl^-]_i$ and especially $[Cl^-]_o$. Despite not knowing the exact change in $[Cl^-]_i$ and $[Cl^-]_o$, the data demonstrate a strong effect of manipulation of astrocytic $[Cl^-]_i$ on neuronal activity has.

Secondly, while the extracellular $[Cl^-]$ is on average around 120 mM, we argue that local differences and Cl^- microdomains exist. The extracellular space is compartmentalized, while the synaptic cleft is rather small with 200-300 nm X 15-25 nm, other region of extracellular space show gaps of 30-60 nm. It has been reported that in hippocampus extracellular $[Cl^-]$ is layer specific and it changes dynamically in an activity dependent manner between 99 and 260 mM²⁰. It has also been shown that the extracellular space shrinks by around 30% during excessive neuronal activity²¹⁻²⁴ and by 60% during changes in state of brain activity²⁵. So large a shrinkage of the extracellular space is expected to increase compartmentalization.

Spatial variance in subcellular distribution of $[Cl^-]_i$ or E_{GABA} was suggested by many reports²⁶⁻³⁶ for which several mechanisms have been proposed^{30,31,37}. A range of $[Cl^-]_i$ from 0-60 mM have been described.

When we change extracellular $[Cl^-]_i$ in cortex by washing in aCSF with different Cl^- concentrations we see massive effects on neuronal excitability. This further supports the assumption that local microdomains can exist. Otherwise, this local change we provoke would have been equalized quickly without having extended effects on the neuronal excitability (Supplemental Figure 5).

The simplest argument for the existence of Cl^- microdomains is the existence of relatively immobile anions. In neurons, the $GABA_A$ -permeant ions Cl^- and HCO_3^- together account for a minority of the intracellular anions. The rest of intracellular anions are amino acids and phosphates³⁸, of which only a small fraction is not associated with macromolecules^{39,40} such as nucleic acid species⁴¹ and proteins⁴². These immobile anionic polymers are not uniformly distributed in the cytoplasm^{43,44}, so it is reasonable to expect that there is a nonuniform distribution of mobile anions to compensate for the distribution of immobile anions also in the extracellular space. Thus, we expect that $[Cl^-]_o$ in the nano-domain of the synaptic cleft is much more dynamic than generally assumed.

We have in response to the reviewer updated the manuscript by adding this important point to the discussion, page 19, line 355-363:

“Collapse of the neuronal Cl⁻ gradient is accelerated by pharmacological inhibition of astrocytic gap junctions in ex vivo brain slices. Furthermore, recordings with Cl⁻ sensitive microelectrodes showed that stimulation of CA3 pyramidal neurones leads to an increase in [Cl⁻]_o, which is reduced by applying a gap junction blocker²⁰. Nevertheless, [Cl⁻]_o in the nano-domain of synaptic clefts is unknown and it can be argued that such extracellular Cl⁻ micro/nanodomains exist. Synaptic isolation of inhibitory synapses by astrocytes is crucial for inhibitory synapse function⁴⁵. Furthermore, immobile anionic polymers are not uniformly distributed in the cytoplasm^{43,44} so it is reasonable to expect that there is a nonuniform distribution of mobile anions to compensate for the distribution of immobile anions also in the extracellular space”

4. GABAAR receptors have been found in Bergmann glia, in the cerebellum. The authors should present and discuss evidence for a significant presence of GABAAR in cortical astrocytes.

We agree with the reviewer and revise the manuscript by adding more convincing references. Page 22, line 447. We analyzed three different sets of transcriptomic data and found evidence for GABA_AR expression in cortical astrocytes as well as in astrocytes outside of cerebellum. Single cell sequencing from astrocytes from cortex and hippocampus detect GABA_AR subunit $\alpha 2$, β and some $\gamma 1$ (<https://portal.brain-map.org/>). The data base by Heintz, published by *Doyle et al*⁴⁶ is based on Ribotrap, sequencing RNA that is translated and bound to the Ribosome. They found the same GABA_AR subunit $\alpha 2$, $\beta 1$ and some $\gamma 1$, as well as $\alpha 4$ to be present in astrocytes. Whole brain RNA sequencing data by Ben Barres identified the same subunits $\alpha 2$, $\beta 1$, $\gamma 1$, and $\alpha 4$ ⁴⁷. In addition to the transcriptomic data it was found that cortical astrocytes in primary culture exhibited rises in [Ca²⁺]_i after stimulation with GABA. The GABA_AR agonist muscimol was also selectively able to elicit rises in [Ca²⁺]_i. At the same time GABA_AR antagonists could block the GABA evoked response⁴⁸. Several other studies found proof for GABA_AR in astrocytes of hippocampus⁴⁹⁻⁵¹. These findings are summarized in Supplemental Table 1.

Supplemental Table 1: Astrocytes express GABA_A receptors^{46,47} (<https://portal.brain-map.org/>).

Allan 10x single cell - whole cortex and hippocampus				Heintz data base summary Doyle et al 2008 Cell Ribotrap Aldh1L1 Astrocytes Cortex Normalized		Ben Barres Brain RNA-Seq Li et al 2018		
Gene	protein name	x378_Astro	x376_Astro	variant	protein name	Mus musculus	Homo sapiens	Homo sapiens - fetal
'Gabra1'	alpha1			465.9412674 BQ268470		0.1±0	0.1±0	0.1±0
				415.8488542 BE945884				
				10.05093654 Z36357				
				5.298788811 Z36357				
'Gabra2'	alpha2	1.9274	3.5516	390.3840247 BB339336		15.2393±3.4944	46.0606±3.6761	0.8997±0.3332
				120.0944862 BQ174589				
				76.47509475 NM_008066				
				8.462597899 AV379247 AV379247 RIKEN full-length enriched, adult				
				6.728615808 AV379247 AV379247 RIKEN full-length enriched, adult				
				5.083145333 BB433285 Transcribed sequences				
'Gabra3'	alpha3			4.906019269 NM_008067		0.1666±0.0536	0.6547±0.0968	2.5734±1.3504
'Gabra4'	alpha4			1628.359824 BB430205		12.4015±0.8353	2.6121±0.5117	0.2129±0.0574
				182.3475983 AK013727				
'Gabra5'	alpha5			32.56755474 BQ175863				
'Gabra6'	alpha6			6.324737669 NM_008068		0.1±0	0.1±0	0.1±0
				5.617565322 AF256198				
'Gabrb1'	beta1	9.9478	11.2202	3051.028922 NM_008069		18.062±3.8636	28.9188±1.9468	3.3455±1.7409
'Gabrb2'	beta2			8.160633955 NM_008070		15.2393±3.4944	46.0606±3.6761	0.8997±0.3332
'Gabrb3'	beta3			159.8195906 BQ175666		0.9581±0.1662	3.4870±0.2149	2.5916±0.4326
				12.76251916 BB367779				
				5.514779964 BB367779				
'Gabrd'	delta			6.461331392 BE946994		0.1±0	0.1±0	0.1±0
				5.260348726 NM_008072				
'Gabre'	epsilon			5.358142582 NM_017369		0.1614±0.0614		
'Gabrg1'	gamma1	0.3147	0.6147	4517.096667 AF156490		37.6960±4.3232	16.882±1.5497	0.6184±0.3926
				322.9320861 AF156490				
				4.765074503 BE946928 Transcribed sequence with weak similarity				
'Gabrg2'	gamma2			15.26860289 AV348821		0.1424±0.0234	0.1905±0.0445	0.0987±0.0012
				4.684561598 AF233778				
'Gabrg3'	gamma3			5.860644796 NM_008074		2.3917±0.1268	0.1003±0.000325	0.1±0
'Gabrp'	pi			6.089978571 BC027245		0.1±0	0.1±0	0.1098±0.0098
				4.116624965 BC027245				
'Gabrq'	theta			5.074559857 NM_020488		0.1±0	0.2520±0.0712	0.4275±0.2106
'Gabbr1'	rho1			5.258867444 NM_008075		0.1±0	0.1±0	0.1±0
'Gabbr2'	rho2			5.317823179 AF024621		0.2917±0.0375	0.1044±0.0044	0.1±0
'Gabbr3'	rho3					0.1±0	0.1017±0.0017	0.1±0
'Aqp4'		10.9980	3.9287	15696.61217 BB193413 Adult male spinal cord cDNA, RIKEN full-len		317.6387±15.0845	449.9831±36.4463	14.0190±6.3434
				823.2821282 U48399				
				113.8031517 AW489155				

5. Fig. 4 data require further clarification and controls.

We thank the reviewer for the following feedback and addressed the single points of critique in detail below:

(i) Muscimol application will activate GABAARs at multiple cell targets, including principal neurons and interneurons, which in turn could affect astrocyte activity. This makes it difficult to interpret the data as a direct effect of astrocyte activity.

We agree with the reviewer that muscimol as well as GABA target all GABA_AR without being specific to any cell type. This is a common problem to all drug application. Unfortunately, no specific pharmacology to selectively address GABA_AR in astrocytes is available. While it is not possible *in vivo* to address astrocytes selectively, it was found that cortical astrocytes in primary culture exhibited rises in [Ca²⁺]_i after stimulation with GABA. The GABA_AR agonist muscimol was also selectively able to elicit rises in [Ca²⁺]_i. At the same time GABA_AR antagonists could block the GABA evoked response⁴⁸.

(ii) Bolus loading with MQAE would label all cells in the region including neuronal elements intermingled with SR101 labelled astroglia. Distinguishing between astroglial and local neuronal MQAE signals would not appear feasible in these conditions.

We agree with the reviewer, MQAE theoretically labels all cells in cortex, while in practice the accumulation into astrocytes is higher compared to neurons. We used 2-PM imaging and the resolution is sufficient to identify cells and identify astrocytes using SR101⁵². Single somata are analysed.

Reviewer #3 (Remarks to the Author):

The manuscript by Unitiet, et al., reports that the astrocytic Cl⁻ is brain state dependent and it regulates cortical neuronal activity. Authors employed very challenging in vivo fiber photometry from awake, behaving mouse brain, imaging intracellular astrocytic Cl⁻. In addition, authors employed NpHR to artificially manipulate the intracellular Cl⁻ upon light stimulation to see if the neuronal activity is altered. Although the authors should be praised for heroic attempt to record astrocytic Cl⁻ and manipulate astrocytic Cl⁻, there are several major concerns that need to be addressed to provide convincing evidence for their claims.

1. In figure 1 authors found astrocytic Cl⁻ is dramatically changed during sensory input or motor activity in behaving mouse brain. Authors used a genetically encoded Cl⁻ sensor mClY, which is a YFP variant that is supposed to be pH-independent. Other predecessor Cl⁻ sensors such as YFP, Clomeleon, and SuperClomeleon all suffer from pH-sensitivity. Authors in this study used YFP as a negative control. Authors found that there was no change with YFP. This negative result with GFAP-YFP is odd because if the positive mClY signal was due to a bona fide Cl⁻, YFP should also give a positive signal for YFP is also a Cl⁻ sensor. This negative data by YFP can be interpreted as the positive signal by mClY is not due to Cl⁻ but rather something other than Cl⁻ or pH.

We agree that majority of genetically encoded biosensors are highly sensitive to pH changes and not suitable to distinguish between chloride and pH changes. We recorded fluorescent signal of mClY on the 2PM via an acute craniotomy while perfusing with aCSF with different pH. This manipulation did not affect that signal significantly in contrast to, for example, KX anesthesia. We added this new data as Supplemental Figure 10. Even though mClY is derived from YFP, the pH interference was removed, photostability was increased and chloride sensitivity was enhanced in validated in an extensive analysis⁵³. The sensor has a K_d for chloride of 14 mM and a pK_a of 5.9 rendering the pH sensitivity for physiological changes in pH rather low. The bleach time constant of mClY is 175 seconds, which is 15-fold greater than YFP. The chloride K_d of YFP at pH 7.5 is 777 mM, which is far away from intracellular chloride and physiological conditions⁵³. While we see changes in mClY and no significant changes in YFP we conclude that the effect is based on Cl⁻ fluctuations and not on pH changes.

Supplemental Figure 10: mClY expressed in vivo astrocytes is pH insensitive.

Fluorescence intensity of mClY expressed in astrocytes was imaged via an acute craniotomy using 2 photon microscopy. aCSF containing different pH was perfused while the fluorescence

intensity of mCIY was imaged. As a control the animal was anesthetized using ketamine xylazine anesthesia. (N = 1 animal). Data represent mean ± SEM.

2. In figure 2D, G, H, authors report a biphasic change in mCIY during movement and whisker stimulation. This biphasic change is reminiscent of the previous report by Woo, et al., 2020, J of Physiology, in which the similar biphasic change of Cl⁻ and its detailed molecular mechanism was described in hippocampus CA1. In particular, the initial increase in mCIY, a putative influx of Cl⁻, is very interesting and this was previously demonstrated to be mediated possibly by GlyT1 (Woo, et al., 2020). However, authors did not pay attention to this initial increase but rather focused on the decreasing phase of mCIY. The decrease of mCIY, a putative efflux of Cl⁻, is also reminiscent of the Cl⁻ efflux reported by Woo, et al, 2020, in which the Cl⁻ efflux was demonstrated to be mediated by Best1 (Woo, et al., 2020). Authors should cite this paper and discuss about the possible molecular mechanism of this biphasic response of mCIY.

We agree with the reviewer, the biphasic response published in Woo et al looks similar to the response we describe. Both changes in [Cl⁻]_i were evoked by neuronal stimulation. The initial increase in [Cl⁻]_i takes around 2 sec and is followed by a prolonged decrease that starts ~7 sec and lasts as long as the stimulation lasts. Woo et al describe a fast rise of 4-6 sec and a slow decay of 70-80 sec. Because we found the fast influx of Cl⁻ was not affected by optogenetic stimulation and that it did not affect neuronal signaling (Figure 3), we focused on the prolonged decrease. Woo et al describe this decrease to be Best-1 mediated. We speculate that GABA_AR are involved in the mechanism. In support of this notion, we found that GABA and muscimol trigger Cl⁻ efflux from astrocytes. Also, Diazepam (a GABA_AR allosteric agonist) causes a decrease in stimulated neuronal responses. The effect is similar to the effect increased baseline astrocytic [Cl⁻]_i has on stimulated neuronal activity (Supplemental Figure 6). This observation points all to a GABA mediated mechanism. We have in response to the reviewer's comment in the discussion added this paragraph. Page 25, line 509-521.

“Instigating neuronal activity by whisker stimulation or movement onset consistently triggered a biphasic response of astrocytic [Cl⁻]_i (Figure 2). Fast increase in astrocytic [Cl⁻]_i could be explained by Cl⁻ influx through, for example, the glycine transporter GlyT1, which is reported to have the same time course as K⁺ uptake through K2P channels⁷. The transporter NKCC1 can be potentially involved too, yet NKCC1 is not expressed in astrocytes in the adult forebrain^{8,9}. Volume changes mediated by increases in extracellular K⁺ are unlikely to trigger a transient increase in [Cl⁻]_i, as astrocytic swelling upon physiological increases of K⁺ develops within much slower time range of minutes¹⁰. Prolonged decrease of astrocytic [Cl⁻]_i is likely mediated by opening of GABA_AR, since puff application of GABA or muscimol triggers a similar Cl⁻ release from astrocytes (Figure 5). Electrical stimulation of Schaffer collateral pathway results in the prolonged decrease in astrocytic [Cl⁻]_i mediated by Best-1 Cl⁻ channel in brain slices⁷, but the observed slow decay (70-80 seconds) is several fold longer than the 10-15 seconds observed in our experiments (Figure 2).”

3. While the authors were focusing on the Cl⁻ efflux phase, they try to mimic the influx of Cl⁻ by employing NpHR, which will increase the intracellular Cl⁻. Thus, this employment of NpHR does not fit very well with their focus on the efflux of Cl⁻ from astrocytes to provide extracellular Cl⁻ for supply to neurons. What the authors really

need is a manipulation that mimics efflux of Cl^- . Therefore, it is unfortunate that there is a disconnect between what they really want to mimic and what they ended up using. We appreciate the reviewer's insightful suggestion and agree that it would be useful to selectively manipulate with GABA_AR in astrocytes. However, the mechanism we hypothesise needs simultaneous activation of GABA_AR in astrocytes and neurons. To address the reviewer's concerns, we first chose to increase the impact of astrocytic chloride, by increasing the baseline $[\text{Cl}^-]_i$. To complement our study on the role of astrocytic $[\text{Cl}^-]_i$ on neuronal activity we added optogenetic manipulation which decreases baseline astrocytic $[\text{Cl}^-]_i$ by new Figure 4. The optogenetic tool SwiChR is a switchable Cl^- channel, which depletes the astrocytic Cl^- gradient. The data shows that during whisker stimulation the astrocytic Cl^- response is diminished. At the same time the neuronal response during the same stimulation protocol increased. This shows that neuronal activity is facilitated when astrocytic Cl^- is reduced. The new data was added in the new Figure 4.

Figure 4: Optogenetic depletion of astrocytic $[\text{Cl}^-]_i$ extends activation-induced neuronal $[\text{Ca}^{2+}]_i$. (a) Experimental protocol: The optogenetic switchable Cl^- channel was expressed in astrocytes to manipulate $[\text{Cl}^-]_i$, while astrocytic $[\text{Cl}^-]_i$ or neuronal $[\text{Ca}^{2+}]_i$ was imaged using mCIY or jRGECO1a, respectively. As the negative control no light or only red-light stimulation was applied to mice expressing SwiChR. The awake, head-fixed mice were whisker stimulated using air puffs. (b) Average projection of Gfap-mCIY and Gfap-SwiChR on the 2PM. (c) Average astrocytic $[\text{Cl}^-]_i$ trace during whisker stimulation in control, light control and upon light-activation of optogenetic tool SwiChR in astrocytes, shading indicates SD. (N = 5 mice). Area under the curve during the second half of whisker stimulation. (P = 0.0278 Paired two-tailed t-test). (d) Average projection of Thy1-GCamp and Gfap-SwiChR on the 2PM and ROIs of single cells that were detected. (e) Raster plot showing the activation profile of the neurons detected in (d). (f) Average neuronal $[\text{Ca}^{2+}]_i$ trace during whisker stimulation in control, light control and upon light-activation of optogenetic tool SwiChR in astrocytes, shading indicates SD. (N = 560 neurons, 6 mice). Ratio between first and second half of neuronal $[\text{Ca}^{2+}]_i$. Paired two-tailed t-test. Data represent mean \pm SEM.

4. In figure 4, authors demonstrate that GABA can induce a decrease in Cl^- in astrocytes. This is a useful addition. However, this needs to be further expanded to address the real question of whether the putative Cl^- efflux in behaving animal during sensory stimulation or motor activity is due to the astrocytic GABA_A receptors. Cell-

type specific manipulation (gene silencing or cKO) of GABA_A receptor subunit in astrocyte is needed.

We thank the reviewer for this valuable suggestion and agree that it would be interesting to silence GABA_AR in astrocytes. However, we believe this is outside the scope of the paper because silencing GABA_AR selectively in astrocytes is not straightforward. GABA_AR are composed of multiple different subunits of which very few or none are specific to either neurons or astrocytes. Therefore, it is highly likely that the loss of a particular subunit will be compensated. Generating and evaluating a gene silencing or cKO model is outside the scope of our paper because we aim to demonstrate that astrocytic Cl⁻ has an impact on neuronal signaling. We added in response to the reviewers comment to the discussion on page 22, line 438-440:

“To confirm a GABA_AR mediated mechanism future studies could employ astrocytic specific manipulation of GABA_AR by either silencing or a conditional knock-out mutant.”

5. Authors claim that astrocytes serve as a source of Cl⁻ required for sustained GABAergic transmission. However, authors do not provide direct evidence for this. Instead, they provide indirect lines of evidence that the increase in Cl⁻ influx by NpHR activation in astrocytes causes a reduced neuronal activity. There is a huge gap between increase in Cl⁻ influx in astrocytes and providing Cl⁻ required for sustained GABAergic transmission in neurons. Authors need to fill in the missing gap with additional evidence. Otherwise, the claim needs to be toned down.

We agree with the reviewer, we are missing direct proof for the hypothesized mechanism. However, our study shows that manipulation of astrocytic [Cl⁻]_i has an impact on neuronal signaling. Also, a similar effect was demonstrated using Diazepam (an allosteric activator or GABA_AR), which supports the suggestion that astrocytic [Cl⁻]_i modulates inhibition by GABA_AR mediated mechanism. The following sentence was added to emphasize that the proposed mechanism is a hypothesis, page 22, line 438-440:

“To confirm a GABA_AR mediated mechanism future studies could employ astrocytic specific manipulation of GABA_AR by either silencing or a conditional knock-out mutant.”

6. Authors claim that NpHR-induced Cl⁻ influx leads to Cl⁻ efflux that increases [Cl⁻]_o. However, authors do not provide evidence for this. The [Cl⁻]_o is already very high, at around 130 mM. It remains unclear how much astrocytic Cl⁻ efflux will increase this already high [Cl⁻]_o, unless authors directly measure the [Cl⁻]_o to demonstrate the increases in [Cl⁻]_o.

The reviewer raised an important point. We want to emphasize that the central and novel finding of our study is that astrocytic [Cl⁻]_i directly affects neuronal activity. While increasing baseline astrocytic [Cl⁻]_i reduces neuronal activity in cortex, decreasing astrocytic [Cl⁻]_i increases neuronal activity during prolonged stimulation. Furthermore, we provide evidence that the underlying mechanism is GABA_AR mediated. We show that GABA triggers Cl⁻ release from astrocytes *in vivo* as well as that an allosteric GABA_AR agonist has similar effects as manipulation astrocytic [Cl⁻]_i. Based on this data we hypothesize a mechanism for our new findings.

We also show that reducing [Cl⁻]_o has a massive effect on neuronal activity, Supplemental Figure 5. In fact, this data demonstrates that Cl⁻ is a rate limiting factor for GABAergic

transmission. As we expect the extracellular space to be compartmentalized (see our response to Reviewer 1 #3), extracellular $[Cl^-]_o$ is likely not uniformly distributed and local gradients might exist. It was shown that $[Cl^-]_o$ changes locally in an activity dependent manner between 99 and 260 mM²⁰. The synaptic cleft is very narrow, and movement even of a few ions has a big impact on the concentration. Another argument is that the collapse of the neuronal Cl^- gradient is accelerated by pharmacological inhibition of astrocytic gap-junctions in *ex vivo* brain slices¹. This suggests that net Cl^- extrusion from the astrocytic network might replenish $[Cl^-]_o$ during intense activation of GABAergic synapses¹. We do not show direct measurements of $[Cl^-]_o$. Unfortunately, there is not sensor available to measure $[Cl^-]_o$. All known Cl^- sensor have a low dynamic range and are not very well-suited differentiation of smaller changes in high chloride concentration (around 120 mM). The same is true for microelectrodes. A good tool to image or record $[Cl^-]_o$ needs to be developed. Furthermore, as $[Cl^-]_o$ in the extracellular space is not evenly distributed the issue of recording *in vivo* Cl^- in the small area of the synaptic cleft needs to be overcome.

7. It is unclear that the collapse of the neuronal Cl^- gradient during intense GABA_A receptor activity is due to an increase in Cl^- accumulation or decrease in $[Cl^-]_o$. Because of very high concentration of 130 mM $[Cl^-]_o$, it is more likely that the collapse is due to an increase in Cl^- accumulation. Authors need to clarify this assumption.

We fully agree with the reviewer. It is not clear whether a reduction of $[Cl^-]_o$ or an increase of neuronal $[Cl^-]_i$ makes the neuronal Cl^- gradient collapse, or both. Therefore, we summarise the literature on this topic in our discussion: page 20-21, line 409-431

*“It is well known that long-lasting activation of inhibitory synapses leads to a significant reduction in the amplitude of IPSPs over time, reflecting a collapse of the Cl^- gradient due to Cl^- influx into neurones through GABA_{AR}^{4-6, 6}. Voltage-clamp recordings from neocortical pyramidal cells and dendrites of hippocampal CA1 pyramidal cell revealed a shift in the reversal potential of GABA_{AR} currents during sustained inhibition indicating a decrease in transmembrane Cl^- gradient^{4,5}. Whole cell patch-clamp recordings in combination with $[Cl^-]_i$ imaging of somata of mature hippocampal CA1 pyramidal neurones, confirmed synaptically activated GABA_{AR} mediated Cl^- accumulation⁶. Collapse of the neuronal Cl^- gradient is accelerated by pharmacological inhibition of astrocytic gap junctions in *ex vivo* brain slices. Furthermore, recordings with Cl^- sensitive microelectrodes showed that stimulation of CA3 pyramidal neurones leads to an increase in $[Cl^-]_o$, which is reduced by applying a gap junction blocker²⁰. Nevertheless, $[Cl^-]_o$ in the nano-domain of synaptic clefts is unknown and it can be argued that such extracellular Cl^- micro/nanodomains exist. Synaptic isolation of inhibitory synapses by astrocytes is crucial for inhibitory synapse function⁴⁵. Furthermore, immobile anionic polymers are not uniformly distributed in the cytoplasm^{43,44} so it is reasonable to expect that there is a nonuniform distribution of mobile anions to compensate for the distribution of immobile anions also in the extracellular space. We here show that manipulating astrocytic $[Cl^-]_i$ by optogenetic stimulation of the Cl^- pump alleviated neuronal activity during locomotion or whisker stimulation resulting in a decrease of plateau and faster recovery of neuronal $[Ca^{2+}]_i$ transients.”*

Minor comments

1. Line 194: accelerates is misspelled

Thanks for finding this typo, it was corrected.

2. Line 199: mice is misspelled

Thanks for pointing this out, it was corrected.

3. Line 245: “lasted for 15.10 ± 10.1” needs a unit.

We thank the reviewer for finding this oversight. The error was corrected.

4. Line 255: that -> than

Thanks for highlighting this, the sentence was changed to make sense.

5. Line 283: “-0.)”?

Thanks for finding this oversight. It was left over from reference formatting and was erased.

We would like to thank the reviewers for giving us the opportunity to strengthen our manuscript with valuable comments and queries. We have worked hard to incorporate the feedback and to address all the reviewers' concerns in a proper way. We believe that our paper has improved considerably. We look forward to hearing from you soon.

Kind regards,

Maiken, Alex and Verena

- 1 Egawa, K., Yamada, J., Furukawa, T., Yanagawa, Y. & Fukuda, A. Cl(-) homeodynamics in gap junction-coupled astrocytic networks on activation of GABAergic synapses. *The Journal of physiology* **591**, 3901-3917 (2013). <https://doi.org/10.1113/jphysiol.2013.257162>
- 2 Kang, J., Jiang, L., Goldman, S. A. & Nedergaard, M. Astrocyte-mediated potentiation of inhibitory synaptic transmission. *Nature neuroscience* **1**, 683-692 (1998). <https://doi.org/10.1038/3684>
- 3 Jiang, L., Sun, S., Nedergaard, M. & Kang, J. Paired-pulse modulation at individual GABAergic synapses in rat hippocampus. *The Journal of physiology* **523 Pt 2**, 425-439 (2000). <https://doi.org/10.1111/j.1469-7793.2000.t01-1-00425.x>
- 4 Staley, K. J. & Proctor, W. R. Modulation of mammalian dendritic GABA(A) receptor function by the kinetics of Cl- and HCO₃- transport. *The Journal of physiology* **519 Pt 3**, 693-712 (1999). <https://doi.org/10.1111/j.1469-7793.1999.0693n.x>
- 5 Staley, K. J., Soldo, B. L. & Proctor, W. R. Ionic mechanisms of neuronal excitation by inhibitory GABA_A receptors. *Science* **269**, 977-981 (1995). <https://doi.org/10.1126/science.7638623>
- 6 Isomura, Y. *et al.* Synaptically activated Cl- accumulation responsible for depolarizing GABAergic responses in mature hippocampal neurons. *Journal of neurophysiology* **90**, 2752-2756 (2003). <https://doi.org/10.1152/jn.00142.2003>
- 7 Woo, J. *et al.* The molecular mechanism of synaptic activity-induced astrocytic volume transient. *The Journal of physiology* **598**, 4555-4572 (2020). <https://doi.org/10.1113/JP279741>
- 8 Plotkin, M. D. *et al.* Expression of the Na(+)-K(+)-2Cl- cotransporter BSC2 in the nervous system. *The American journal of physiology* **272**, C173-183 (1997). <https://doi.org/10.1152/ajpcell.1997.272.1.C173>
- 9 Clayton, G. H., Owens, G. C., Wolff, J. S. & Smith, R. L. Ontogeny of cation-Cl- cotransporter expression in rat neocortex. *Brain research. Developmental brain research* **109**, 281-292 (1998).
- 10 Florence, C. M., Baillie, L. D. & Mulligan, S. J. Dynamic volume changes in astrocytes are an intrinsic phenomenon mediated by bicarbonate ion flux. *PloS one* **7**, e51124 (2012). <https://doi.org/10.1371/journal.pone.0051124>
- 11 Zhao, J., Wang, D. & Wang, J. H. Barrel cortical neurons and astrocytes coordinately respond to an increased whisker stimulus frequency. *Mol Brain* **5**, 12 (2012). <https://doi.org/10.1186/1756-6606-5-12>
- 12 Saper, C. B., Fuller, P. M., Pedersen, N. P., Lu, J. & Scammell, T. E. Sleep state switching. *Neuron* **68**, 1023-1042 (2010). <https://doi.org/10.1016/j.neuron.2010.11.032>
- 13 Aston-Jones, G. & Bloom, F. E. Activity of norepinephrine-containing locus coeruleus neurons in behaving rats anticipates fluctuations in the sleep-waking cycle. *The Journal of neuroscience : the official journal of the Society for Neuroscience* **1**, 876-886 (1981).
- 14 Kjaerby, C. *et al.* Memory-enhancing properties of sleep depend on the oscillatory amplitude of norepinephrine. *Nat Neurosci* **25**, 1059-1070 (2022). <https://doi.org/10.1038/s41593-022-01102-9>
- 15 Weiling, N. L. *et al.* KCC2 drives chloride microdomain formation in dendritic blebbing. *Cell Rep* **41**, 111556 (2022). <https://doi.org/10.1016/j.celrep.2022.111556>
- 16 Gensch, T., Untiet, V., Franzen, A., Kovermann, P. & Fahlke, C. in *Advanced Time-Correlated Single Photon Counting Applications* Vol. 111 *Springer Series in Chemical Physics* (ed Wolfgang Becker) Ch. 4, 189-211 (Springer International Publishing, 2015).
- 17 Kaneko, H., Putzier, V., Frings, S. & Gensch, T. in *Calcium-Activated Chloride Channels* Vol. 53 *Current Topics in Membranes* (ed C. M. Fuller) 167-+ (Elsevier Academic Press Inc, 2002).
- 18 Verkman, A. S., Sellers, M. C., Chao, A. C., Leung, T. & Ketcham, R. Synthesis and characterization of improved chloride-sensitive fluorescent indicators for biological applications. *Analytical biochemistry* **178**, 355-361 (1989).
- 19 Meyer, H. S. *et al.* Inhibitory interneurons in a cortical column form hot zones of inhibition in layers 2 and 5A. *Proceedings of the National Academy of Sciences of the United States of America* **108**, 16807-16812 (2011). <https://doi.org/10.1073/pnas.1113648108>

- 20 Kroeger, D., Tamburri, A., Amzica, F. & Sik, A. Activity-dependent layer-specific changes in the extracellular chloride concentration and chloride driving force in the rat hippocampus. *Journal of neurophysiology* **103**, 1905-1914 (2010). <https://doi.org/10.1152/jn.00497.2009>
- 21 Dietzel, I., Heinemann, U., Hofmeier, G. & Lux, H. D. Transient changes in the size of the extracellular space in the sensorimotor cortex of cats in relation to stimulus-induced changes in potassium concentration. *Experimental brain research* **40**, 432-439 (1980).
- 22 Freygang, W. H., Jr. & Landau, W. M. Some relations between resistivity and electrical activity in the cerebral cortex of the cat. *J Cell Comp Physiol* **45**, 377-392 (1955). <https://doi.org/10.1002/jcp.1030450305>
- 23 McBain, C. J., Traynelis, S. F. & Dingledine, R. Regional variation of extracellular space in the hippocampus. *Science* **249**, 674-677 (1990). <https://doi.org/10.1126/science.2382142>
- 24 Van Harrevelde, A. & Khattab, F. I. Changes in cortical extracellular space during spreading depression investigated with the electron microscope. *Journal of neurophysiology* **30**, 911-929 (1967). <https://doi.org/10.1152/jn.1967.30.4.911>
- 25 Xie, L. *et al.* Sleep drives metabolite clearance from the adult brain. *Science* **342**, 373-377 (2013). <https://doi.org/10.1126/science.1241224>
- 26 Pouzat, C. & Marty, A. Somatic recording of GABAergic autoreceptor current in cerebellar stellate and basket cells. *The Journal of neuroscience : the official journal of the Society for Neuroscience* **19**, 1675-1690 (1999). <https://doi.org/10.1523/JNEUROSCI.19-05-01675.1999>
- 27 Astorga, G. *et al.* An excitatory GABA loop operating in vivo. *Frontiers in cellular neuroscience* **9**, 275 (2015). <https://doi.org/10.3389/fncel.2015.00275>
- 28 Berglund, K. *et al.* Imaging synaptic inhibition in transgenic mice expressing the chloride indicator, Clomeleon. *Brain Cell Biol* **35**, 207-228 (2006). <https://doi.org/10.1007/s11068-008-9019-6>
- 29 Duebel, J. *et al.* Two-photon imaging reveals somatodendritic chloride gradient in retinal ON-type bipolar cells expressing the biosensor Clomeleon. *Neuron* **49**, 81-94 (2006). <https://doi.org/10.1016/j.neuron.2005.10.035>
- 30 Foldy, C., Lee, S. H., Morgan, R. J. & Soltesz, I. Regulation of fast-spiking basket cell synapses by the chloride channel ClC-2. *Nat Neurosci* **13**, 1047-1049 (2010). <https://doi.org/10.1038/nn.2609>
- 31 Glykys, J. *et al.* Local impermeant anions establish the neuronal chloride concentration. *Science* **343**, 670-675 (2014). <https://doi.org/10.1126/science.1245423>
- 32 Romo-Parra, H., Trevino, M., Heinemann, U. & Gutierrez, R. GABA actions in hippocampal area CA3 during postnatal development: differential shift from depolarizing to hyperpolarizing in somatic and dendritic compartments. *Journal of neurophysiology* **99**, 1523-1534 (2008). <https://doi.org/10.1152/jn.01074.2007>
- 33 Schmidt, T. *et al.* Differential regulation of chloride homeostasis and GABAergic transmission in the thalamus. *Scientific reports* **8**, 13929 (2018). <https://doi.org/10.1038/s41598-018-31762-2>
- 34 Szabadics, J. *et al.* Excitatory effect of GABAergic axo-axonic cells in cortical microcircuits. *Science* **311**, 233-235 (2006). <https://doi.org/10.1126/science.1121325>
- 35 Zorrilla de San Martin, J., Trigo, F. F. & Kawaguchi, S. Y. Axonal GABA(A) receptors depolarize presynaptic terminals and facilitate transmitter release in cerebellar Purkinje cells. *The Journal of physiology* **595**, 7477-7493 (2017). <https://doi.org/10.1113/JP275369>
- 36 Untiet, V. *et al.* Elevated Cytosolic Cl⁻ Concentrations in Dendritic Knobs of Mouse Vomeronasal Sensory Neurons. *Chemical senses* **41**, 669-676 (2016). <https://doi.org/10.1093/chemse/bjw077>
- 37 Khirug, S. *et al.* GABAergic depolarization of the axon initial segment in cortical principal neurons is caused by the Na-K-2Cl cotransporter NKCC1. *The Journal of neuroscience : the official journal of the Society for Neuroscience* **28**, 4635-4639 (2008). <https://doi.org/10.1523/JNEUROSCI.0908-08.2008>
- 38 Morawski, M. *et al.* Ion exchanger in the brain: Quantitative analysis of perineuronally fixed anionic binding sites suggests diffusion barriers with ion sorting properties. *Scientific reports* **5**, 16471 (2015). <https://doi.org/10.1038/srep16471>

- 39 Masuda, T., Dobson, G. P. & Veech, R. L. The Gibbs-Donnan near-equilibrium system of heart. *The Journal of biological chemistry* **265**, 20321-20334 (1990).
- 40 Veech, R. L., Kashiwaya, Y., Gates, D. N., King, M. T. & Clarke, K. The energetics of ion distribution: the origin of the resting electric potential of cells. *IUBMB Life* **54**, 241-252 (2002). <https://doi.org/10.1080/15216540215678>
- 41 Manning, G. S. The molecular theory of polyelectrolyte solutions with applications to the electrostatic properties of polynucleotides. *Q Rev Biophys* **11**, 179-246 (1978). <https://doi.org/10.1017/s0033583500002031>
- 42 Gianazza, E. R., G. P. Size and charge distribution of macromolecules in living systems. *Journal of Chromatography A* **193**, 1-8 (1980). [https://doi.org/10.1016/S0021-9673\(00\)81438-7](https://doi.org/10.1016/S0021-9673(00)81438-7)
- 43 Chen, X., Wu, X., Wu, H. & Zhang, M. Phase separation at the synapse. *Nat Neurosci* **23**, 301-310 (2020). <https://doi.org/10.1038/s41593-019-0579-9>
- 44 Gut, G., Herrmann, M. D. & Pelkmans, L. Multiplexed protein maps link subcellular organization to cellular states. *Science* **361** (2018). <https://doi.org/10.1126/science.aar7042>
- 45 Takano, T. *et al.* Chemico-genetic discovery of astrocytic control of inhibition in vivo. *Nature* **588**, 296-302 (2020). <https://doi.org/10.1038/s41586-020-2926-0>
- 46 Doyle, J. P. *et al.* Application of a translational profiling approach for the comparative analysis of CNS cell types. *Cell* **135**, 749-762 (2008). <https://doi.org/10.1016/j.cell.2008.10.029>
- 47 Li, Q. *et al.* Developmental Heterogeneity of Microglia and Brain Myeloid Cells Revealed by Deep Single-Cell RNA Sequencing. *Neuron* **101**, 207-223 e210 (2019). <https://doi.org/10.1016/j.neuron.2018.12.006>
- 48 Nilsson, M., Eriksson, P. S., Ronnback, L. & Hansson, E. GABA induces Ca²⁺ transients in astrocytes. *Neuroscience* **54**, 605-614 (1993). [https://doi.org/10.1016/0306-4522\(93\)90232-5](https://doi.org/10.1016/0306-4522(93)90232-5)
- 49 Fraser, D. D. *et al.* GABAA/benzodiazepine receptors in acutely isolated hippocampal astrocytes. *The Journal of neuroscience : the official journal of the Society for Neuroscience* **15**, 2720-2732 (1995).
- 50 Kettenmann, H., Backus, K. H. & Schachner, M. gamma-Aminobutyric acid opens Cl⁻ channels in cultured astrocytes. *Brain research* **404**, 1-9 (1987).
- 51 MacVicar, B. A., Tse, F. W., Crichton, S. A. & Kettenmann, H. GABA-activated Cl⁻ channels in astrocytes of hippocampal slices. *The Journal of neuroscience : the official journal of the Society for Neuroscience* **9**, 3577-3583 (1989).
- 52 Nimmerjahn, A. & Helmchen, F. In vivo labeling of cortical astrocytes with sulforhodamine 101 (SR101). *Cold Spring Harbor protocols* **2012**, 326-334 (2012). <https://doi.org/10.1101/pdb.prot068155>
- 53 Zhong, S., Navaratnam, D. & Santos-Sacchi, J. A genetically-encoded YFP sensor with enhanced chloride sensitivity, photostability and reduced pH interference demonstrates augmented transmembrane chloride movement by gerbil prestin (SLC26a5). *PloS one* **9**, e99095 (2014). <https://doi.org/10.1371/journal.pone.0099095>

REVIEWERS' COMMENTS

Reviewer #1 (Remarks to the Author):

Here the authors present a revised manuscript that is much improved over the previous version. The authors have addressed most of my concerns. The author's argument was strengthened by the addition of experiments that reduce astrocyte $[Cl^-]_i$ using SwiChR and electrophysiologically recorded IPSCs. There are a few minor (but important) concerns that need to be revised, however.

- 1) Supplemental Figure 8. The left panel in d and the middle panel in e appear to be the same image. Probably the image of d is an oversight. This is a serious mistake and should be corrected.
- 2) I always misinterpret figure 3. The NpHR should increase astrocyte $[Cl^-]_i$, however, figure 3b shows a decrease in astrocyte $[Cl^-]_i$, which is highly misleading. This may be due to the fact the definition of time 0 is ambiguous; time 0 is often misinterpreted as the onset of optogenetics stimulation. Please revise it.
- 3) Supplemental Figure 9. Line 652. "Fluoresce" should be "fluorescence."

Reviewer #2 (Remarks to the Author):

The authors have made a significant effort to address the comments in full. My remaining concerns are relatively minor, as they are mainly about interpretation.

1. The authors attempt a mechanistic interpretation of how the changes in astrocyte Cl^- might affect ion homeostasis in neurons. Their key argument is that extracellular Cl^- is somehow compartmentalized. This is not a plausible assumption, as far as the free Cl^- ions (completely dissociated, strong electrolyte condition, ~ 1 nm hydration radius, ~ 1 nm Debye's length) are concerned. Diffusivity of Cl^- , which is around $1 \mu m^2/ms$, is too high to allow significant compartmentalization in the absence of strong continuous local Cl^- sources (hotspots). The argument about the heterogeneous distribution of chloride in the cell cytoplasm does not stand as we must consider free, rather than bound, ions that account for the Donnan's equilibrium. The double ion layers next to 'immobile anions', charged elements of the cytoskeleton, charged membranes of various cell organelles, etc. have nothing to do with the latter. The argumentation in some neuroscience papers referring to EGABA in this context, unfortunately, reflects the lack of basic knowledge of electrochemistry.

This, however, does not detract from the potential importance of the present findings. This Reviewer simply suggests to omit the highly questionable interpretation, admitting instead the uncertainty about the mechanism. Admitting limitations and uncertainty only strengthens intellectual standing of the paper.

2. The authors cite some published data on GABA-induced Ca^{2+} entry in astrocytes as evidence for astroglial GABAARs, but the phenomenon has been related mainly to the activation of astroglial GABA transporters GAT-1. Again, there is nothing wrong in saying that there has been some, albeit incomplete, evidence for GABAARs in cortical astroglia, etc.

3. It is important to stress in the text that SR101 was used to focus MQAE imaging on the astrocyte somata, as explained by the authors, to exclude contamination from neuronal signalling.

Reviewer #3 (Remarks to the Author):

After the first round of revision, authors tried in good faith to address the comments that I have raised. Most of my concerns were well-addressed. However, one concern was not fully addressed.

1. Authors continue to claim that astrocytic GABA_A is mediating the Cl^- efflux in vivo, although they acknowledge that they do not provide enough evidence for this claim. The only piece of

evidence to support the claim comes from the observation that muscimol application causes an astrocytic $[Cl^-]_i$ decrease. This piece of evidence is obviously not enough to support their claim (and other reviewers also point this out). To provide strong and convincing evidence, authors need to perform astrocyte-specific gene-silencing (or conditional knockout) of one of the GABA α subunits. Without this set of data, authors need to tone down their claim regarding the contribution of astrocytic GABA α .

2. There are many typos and spelling errors that authors need to fix. For example, "knock out" is misspelled as "know out".

REVIEWERS' COMMENTS

Reviewer #1 (Remarks to the Author):

Here the authors present a revised manuscript that is much improved over the previous version. The authors have addressed most of my concerns. The author's argument was strengthened by the addition of experiments that reduce astrocyte $[Cl^-]_i$ using SwiChR and electrophysiologically recorded IPSCs. There are a few minor (but important) concerns that need to revise, however.

1) Supplemental Figure 8. The left panel in d and the middle panel in e appear to be the same image. Probably the image of d is an oversight. This is a serious mistake and should be corrected.

We deeply apologize for this oversight. The image got duplicated during image editing. The mistake is fixed, and the correct image is displayed.

2) I always misinterpret figure 3. The NpHR should increase astrocyte $[Cl^-]_i$, however, figure 3b shows a decrease in astrocyte $[Cl^-]_i$, which is highly misleading. This may be due to the fact the definition of time 0 is ambiguous; time 0 is often misinterpreted as the onset of optogenetics stimulation. Please revise it.

Thanks for pointing this out. The optogenetic stimulation of NpHR indeed increases astrocytic $[Cl^-]_i$, as shown in Suppl figures 3 and 4. Figure 3b, however demonstrates astrocytic $[Cl^-]_i$ transients in response to transition from stationary to mobile recorded either in control or during constant stimulation of NpHR; when NpHR is active $[Cl^-]_i$ transient is smaller. We modified the figure by adding a schematic illustration of the protocol, to make it clearer. Furthermore, the optogenetic stimulation is labeled in the representative traces in Figure 3b, d, g, and i.

3) Supplemental Figure 9. Line 652. “Fluoresce” should be “fluorescence.”

Our reply: Thanks for pointing out this typo. The error has been fixed.

Reviewer #2 (Remarks to the Author):

The authors have made a significant effort to address the comments in full. My remaining concerns are relatively minor, as they are mainly about interpretation.

1. The authors attempt a mechanistic interpretation of how the changes in astrocyte Cl^- might affect ion homeostasis in neurons. Their key argument is that extracellular Cl^- is somehow compartmentalized. This is not a plausible assumption, as far as the free Cl^- ions (completely dissociated, strong electrolyte condition, ~ 1 nm hydration radius, ~ 1 nm Debye's length) are concerned. Diffusivity of Cl^- , which is around $1 \mu m^2/ms$, is too high to allow significant compartmentalization in the absence of strong continuous local Cl^- sources (hotspots). The argument about the heterogeneous distribution of chloride in the cell cytoplasm does not stand as we must consider free, rather than bound, ions that account for the Donnan's equilibrium. The double ion layers next to 'immobile anions', charged elements of the cytoskeleton, charged membranes of various cell organelles, etc. have nothing to do with the latter. The argumentation

in some neuroscience papers referring to EGABA in this context, unfortunately, reflects the lack of basic knowledge of electrochemistry.

This, however, does not detract from the potential importance of the present findings. This Reviewer simply suggests to omit the highly questionable interpretation, admitting instead the uncertainty about the mechanism. Admitting limitations and uncertainty only strengthens intellectual standing of the paper.

We removed the interpretation in question:

~~“A recent publication shows that the equilibrium potential of GABA_ARs in pyramidal neurones is dependent on the time of day, being hyperpolarised during sleep and becoming more depolarised during wakefulness⁵⁹. Reversal potential of GABA_AR (E_{GABAA}) depends on neuronal transmembrane chloride gradient defined by [Cl⁻]_i and [Cl⁻]_o. The study also shows that following sleep deprivation expression of NKCC1, and hence Cl⁻ accumulation into neurones, is upregulated and that this effect can be reversed by blocking NKCC1 using bumetanide⁵⁹. At the same time E_{GABAA} also depends on [Cl⁻]_o in the synaptic cleft regulated by astrocytic [Cl⁻]_i. High astrocytic [Cl⁻]_i during sleep provides Cl⁻ to maintain a E_{GABAA} at hyperpolarised levels, whereas lower [Cl⁻]_i during wakefulness restricts Cl⁻ availability and results in depolarising shift of E_{GABAA}. Hence, astrocytic [Cl⁻]_i participates in modulating neuronal E_{GABAA} in a brain state dependent manner and must be taking into consideration.”~~

2. The authors cite some published data on GABA-induced Ca²⁺ entry in astrocytes as evidence for astroglial GABAARs, but the phenomenon has been related mainly to the activation of astroglial GABA transporters GAT-1. Again, there is nothing wrong in saying that there has been some, albeit incomplete, evidence for GABAARs in cortical astroglia, etc.

We are somewhat confused about this comment: we do not discuss Ca²⁺ entry to astrocytes; as indeed such an entry is secondary to GAT activation or depolarisation. There are compelling evidence for expression of GABA_A receptors in hippocampal protoplasmic astrocytes and in cerebellar astroglia; data from cortex are less complete and this we now indicate in the text as suggested.

Page 11, line 259-262:

“Protoplasmic astrocytes in different brain regions, including cortex, are known to express GABA_AR^{40-43,47-50}. It is conceivable to hypothesise that activation of astrocytic GABA_ARs generates Cl⁻ efflux that increases [Cl⁻]_o thus strengthening inhibitory transmission during prolonged episodes of neuronal activity.”

3. It is important to stress in the text that SR101 was used to focus MQAE imaging on the astrocyte somata, as explained by the authors, to exclude contamination from neuronal signalling.

We modified the manuscript accordingly:

Page 10, line 225-228:

“Astrocytes loaded with the Cl⁻ dye MQAE were imaged while cell type identity was confirmed by co-labelling with astrocyte-specific dye SR-101^{45,46} which allowed to focus on astrocytes to exclude contamination from neuronal signalling (Figure 5f).”

Reviewer #3 (Remarks to the Author):

After the first round of revision, authors tried in good faith to address the comments that I have raised. Most of my concerns were well-addressed. However, one concern was not fully addressed.

1. Authors continue to claim that astrocytic GABA_A is mediating the Cl⁻ efflux in vivo, although they acknowledge that they do not provide enough evidence for this claim. The only piece of evidence to support the claim comes from the observation that muscimol application causes an astrocytic [Cl⁻]_i decrease. This piece of evidence is obviously not enough to support their claim (and other reviewers also point this out). To provide strong and convincing evidence, authors need to perform astrocyte-specific gene-silencing (or conditional knockout) of one of the GABA_A subunits. Without this set of data, authors need to tone down their claim regarding the contribution of astrocytic GABA_A.

We changed the text as suggested.

Page 14-15, line 344-346:

“There are however other possible molecular pathways which may mediate Cl efflux from astrocytes, including opening of anion channels such as bestrophin-1 (Best-1)⁶⁰ or pannexin-1 (Panx1)⁶⁴.”

Page 15, line 354-355:

“Additional experiments with specific silencing of various channels are needed to fully resolve this question.”

2. There are many typos and spelling errors that authors need to fix. For example, "knock out" is misspelled as "know out".

We carefully proofread the paper.